# A qualitative study to understand public views on the relative value of health gains for children and young people in Australia compared to adults

Ashwini De Silva[1]*, Cate Bailey[1], Nancy Devlin[1], Richard Norman[2], Tianxin Pan[1], Tessa Peasgood[1,3], on behalf of the QUality Of life in Kids: Key evidence to strengthen decisions in Australia (QUOKKA) project team¶

**1** Melbourne Health Economics, Centre for Health Policy, Melbourne School of Population and Global Health, University of Melbourne, Melbourne, Australia, **2** School of Population Health, Curtin University, Perth, Australia **3** Division of Population Health, School of Medicine and Population Health, University of Sheffield, Sheffield, United Kingdom

¶ Complete list of investigators of the QUOKKA project team can be found in the acknowledgments section.
* ashwini.desilva@student.unimelb.edu.au

## Abstract

### Objectives

Standard economic evaluation methods assume that quality-adjusted life years (QALYs) have equal social value, regardless of recipient. However, evidence suggests that people place greater social value on health gains for children. This study examines the factors driving age-related preferences for health gains.

### Methods

Think-aloud, semi-structured interviews were conducted with Australian adolescents (n = 7), non-parents (n = 11), parents with healthy children (n = 8) and parents of children with health conditions (n = 15) over a period of four months (27th March 2023–20th July 2023). Participants completed Person Trade-Off (PTO) and attitudinal questions about resource allocation for improvements in life extension, mental health, mobility, and pain/discomfort choosing between interventions for adults (ages 40 or 55) and younger people (ages one month to 24). Thematic analysis was employed to identify fundamental reasoning patterns.

### Results

Nine themes emerged, illustrating participants' complex reasoning. They considered differences in the impact of health problems at various ages, with difficulty envisaging mental health impacts for very young children. Emotional responses were strongest around children in pain. Adolescents tended to prioritize younger people, while parents often emphasized adults' caregiving role. Most participants prioritized based on

**Data availability statement:** The data in this study consists of quotes extracted from participant transcripts. These quotes are included within the manuscript and the Supporting Information (S3 Table). Consent from participants was obtained specifically for the use of these quotes in the study. However, full transcripts are not made publicly available as participant consent was not obtained to share the complete transcripts with the public. The data included currently within the manuscript and the Supporting Information Files (Quotes from participants, the coding tree, demographic characteristics) would constitute the "minimal data set".

**Funding:** Cate Bailey, Nancy Devlin, Richard Norman and Tessa Peasgood- Medical Research Future Fund (MRFF), Department of Health and Aged Care, Australian Government (APP1200816) https://www.health.gov.au/our-work/medical-research-future-fund, Ashwini De Silva- EuroQol Research Foundation grant number 348-PHD https://euroqol.org/ The funders did not and will not have a role in study design, data collection and analysis, decision to publish, or preparation of the manuscript.

**Competing interests:** I have read the journal's policy and the authors of this manuscript have the following competing interests: [ ND, RN, TPA and TPE are all members of the EuroQol group. This does not alter our adherence to PLOS one policies on sharing data and materials].

age in PTO questions, though some adults objected to prioritizing healthcare based on age.

## Conclusion

Choices were shaped by perceptions of the impact of the health states. These qualitative insights help to inform the development of different approaches in healthcare resource allocation highlighting the importance of involving a diverse range of participants with varying views in the decision-making process. The findings also provide insight into interpreting quantitative results from PTO tasks.

## 1. Introduction

Quality-Adjusted Life Years (QALYs) are commonly used as an outcome measure in economic evaluation where they are usually given equal value regardless of who receives them [1]. However, in principle, there could be different societal value of health gain depending on the recipient [2].

Differences in the age-related social values of QALYs are usually based on one of two rationales, efficiency considerations or equity/fairness considerations [3]. Efficiency considerations draw on the relationship between economic productivity and age, with consideration for broader social contributions to vary by age [4]. Fairness considerations draw on the fair innings argument. This was first presented by Harris [5] who argues that everyone should be given an equal opportunity to reach a normal span of years. According to this view, those who have not had their fair innings need to be prioritized. Williams [6] extended the fair innings argument to incorporate a person's quality of life as well as length of life. Tsuchiya et al. [7] reported that people favour giving priority to younger people based on the fair innings argument, and to older people based on efficiency considerations. Schwappach [8] suggested the social value of a QALY may vary according to patients' characteristics, i.e., age, social role, lifestyle or severity of illness.

Existing studies have explored whether age should be considered as a criterion in allocating healthcare resources but the evidence on preference relating to age-related prioritization is mixed. Some studies identified a willingness to prioritize all children aged below 15 [9]. Other studies suggest support for prioritising children but only provide evidence for those older than 5 years. For example, Richardson et al. [10] reported age weights for ages from 5 to 70. They reported a preference for age 5, 10, 15 and 20 for life extension and age 5, 10, 15 for quality of life improvements. Petrou et al. [11] reported relative age weights from a person trade-off (PTO) study and identified a preference for prioritizing the younger age for life extending treatments. In contrast, there is also a study which found participants prioritize adults aged 40 and 70 years over children aged 10 years [12]. Some studies produce findings suggesting everyone should be equally treated [13]. A few qualitative studies analyzed people's views on prioritizing treatments for children compared to adults. Aidem [14] reported that policy makers believe healthcare needs to be prioritized

based on efficiency and equity. Kuder and Roeder [15] reported that people believe patients should not be treated differently based on their age.

A recent systematic review synthesised international evidence on the relative social value of health gains for children (generally referring to individuals under 18 years of age, unless otherwise specified) and those of adults [16]. The review found evidence that the public were willing to prioritize children's health gains over adult's health gains. However, the review identified variations in results (1) based on the study methodology. For example the review identified differences in results based on the type of question, i.e., attitudinal questions [17,18] compared to choice based numerical questions [19], (2) across different perspectives the study questions were framed, i.e., prioritization within the family, or as a citizen or adopting a decision maker perspective, (3) based on the age of the child, (4) based on participant characteristics such as age, gender, parental status and (5) based on whether the health gain referred to extensions of length of life or improvements in quality of life.

One of the limitations identified in this review was the limited number of studies that explored the rationale behind participant choices. Tsuchiya [20] is a rare example. Therefore, further research is needed to understand what drives individuals' responses to preference elicitation questions in which age of the recipient of health gain differs. Qualitative work can help interpret such variations in findings, by providing an understanding of the underlying reasons, and insights into participants' thinking patterns and principles when responding to different types of questions [14,15].

A range of methods are available to elicit public preferences regarding age related prioritization. PTO is widely used to estimate social value weights for health gains in the context of health state valuation and to estimate social value across different groups and treatment characteristics [11,17]. Using PTO to elicit preferences towards treating patients of different ages entails asking participants to make choices between pairs of hypothetical health programs that benefit patients from different age categories [17,21].

This current study is a part of the QUality Of life in Kids: Key evidence to strengthen decisions in Australia (QUOKKA) project and forms the qualitative component of a mixed method PTO study to estimate the average relative weight for health gains for children and young people (aged 0–24 years) compared to health gains for adults [22]. The quantitative analysis of the mixed method PTO [23] reported differences in willingness to prioritize children for healthcare interventions. A sub sample completed one-to-one interviews whilst completing the main PTO survey. This paper focuses on the qualitative analysis of these interviews.

The overall aim of this study is to provide evidence to decision makers in Australia on public opinion regarding the social value of child health gains relative to adult health gains. To achieve this aim, the qualitative analysis of interviews focused on five objectives: (1) understand how participants interpret and make choices in the PTO and what information they focus on, (2) explore whether they think the PTO questions can identify the relative weight they would give to improving child versus adult health, (3) understand participants' reasons behind their preferences, (4) understand the reasons behind any inconsistencies between attitudinal questions and PTO responses and (5) understand how strongly views are held through subjecting participants' opinions and responses to scrutiny, alternative views and disagreement.

## 2. Methods

### 2.1 Recruitment and participants

Existing literature has found that attitudes towards prioritizing child health gains vary depending on the age and parenthood status of the participants [16]. In addition to age and parenthood, we hypothesized that parent's experiences of child ill health may be relevant. Therefore, our recruitment ensured coverage of non-parents, parents of healthy children, and parents of children with a health condition across different age groups.

We also included older adolescents (aged 16–18 years) at the request of the QUOKKA's Decision Makers' Panel. The recent review [16] identified that there was a lack of qualitative studies specifically examining adolescent's perspective on age-based healthcare prioritization, yet studies have shown it is feasible for adolescents to value health states [24] which

 

involves tasks with a similar level of cognitive and emotional difficulty to PTO questions. Adolescents (aged 16–18) coped well during the pilot interviews, which are described at length elsewhere [22].

We anticipated saturation would be reached at a sample of 40 interviews on the basis of Ritchie et al. [25] who suggested that studies involving a very diverse population might require an increased sample size, but a sample of fewer than 50 will be adequate for individual interviews. This is further supported by Hennink et al. [26] who suggested saturation would be achieved between 16–24 interviews and Guest et al. [27] who found saturation occurred within 12 interviews. Consideration of saturation adopted the approach by Guest et al. [27] which "refers to the point during data analysis at which incoming data points (interviews) produce little or no new useful information relative to the study objectives" [27].

Participants were recruited through two mechanisms to ensure inclusion of a breadth of perspectives. The first sample of participants mainly focused on adolescents and adults without children and was recruited through a commercial company, CRNRSTONE. CRNRSTONE invited participants on their panel who had expressed an interest in taking part in qualitative research. The second sample focused on parents of children who had experienced a health issue at some point and was recruited from participants in the QUOKKA Pediatric Multi-Instrument Comparison Study Protocol (P-MIC) study [28] who had previously consented to be contacted for future research. Parents of healthy children were included in both the samples. The P-MIC study included 1000 parents or caregivers of Australian children and adolescents aged 2−18 who attended the Royal Children's Hospital (RCH). The RCH is the largest comprehensive children's hospital in Victoria, Australia. It covers a broad range of health conditions [28]. The two different recruitment approaches ensured coverage across our desired target sample. Ethics approval was received from the University of Melbourne human ethics committee [Reference number: 2023-24869-47516-7]. Informed consent was obtained via a self-complete online questionnaire.

## 2.2 Survey design

The survey included six components, including consent and introduction video, seven PTO questions, feedback questions on comprehension, questions asking for reasons for PTO answers, attitudinal questions on health prioritization and demographic questions. Further details of the tasks and questions are provided in the published protocol study [22] and the paper discussing the quantitative results [23]. The seven PTO tasks involved different aspects of health improvement, including life extension (2 or 5 years), and improvements in aspects of quality of life (mental health, mobility and pain or discomfort). Examples are shown in Figure 1 and 2. There were four life extension questions and three quality of life questions.

Participants were asked to make choices between pairs of interventions, one impacting an adult group (either 40 or 55 years old) and the other younger group consisting of 13 age categories (one month, even number of years between 1 year and 24 years old). These age categories were chosen based on a recent systematic review [16] that identified evidence gaps in very young children (<5 years) and young adults (18–24 years) in studies exploring social value. One of the life extension questions (applied to all participants) compared young people to other young people of a different age as part of a chaining test for the quantitative study. The ages used in the PTO questions was randomly selected, however for the final four interviews an age < 4 years was chosen for the younger age group to further explore findings arising from the analysis of the main survey data. Half of the sample were randomly given the option to select 'no preference' (unforced-arm) between the two hypothetical health programs in the seven PTO tasks, the other half were always required to choose between Program A or B (forced-arm). Interviewer prompts included discussion of their likely answer if they had seen the alternative presentation (i.e., when a choice between two programs was forced).

## 2.3 Data collection

Interviews drew on a combination of 'think-aloud' and interviewer prompts. The direct verbalization of thoughts as participants answered PTO and attitudinal questions aimed to capture their cognitive process [29] and the probing interviewer

Programs A and B are **treatments that will improve patients' health temporarily**.

- If they receive the treatment patients will not experience any loss in health at all.
- If they do not get the treatment, they will experience a 2-year physical health illness with the symptom pain after which they will return to normal health with no long-term health consequence.

The only difference between these groups is the age of the patients.

Remember that we want you to **assume that these are the same in both programs**.

- Overall costs
- The carers' health and wellbeing
- Any loss of income of carers or patients

We want you to tell us which program the decision makers should choose.

| **Program A** | **Program B** |
|---|---|
| Age of patients: **2** | Age of patients: **40** |
| Prevents a 2-year illness which has the symptom **pain** after which they would return to normal health with no long-term health consequence. | Prevents a 2-year illness which has the symptom **pain** after which they would return to normal health with no long-term health consequence. |
| **100 patients** treated | **100 patients** treated |

**Fig 1. Quality of Life question example.**

Programs A and B are **life extending treatments** that will make patients live longer, without which they will die now.

If they receive the program patients will get an **extra 2 years of life spent in good health** after which they will die.

The only difference between these groups is the age of the patients.

Remember that we want you to **assume that these are the same in both programs**.

- Overall costs
- The carers' health and wellbeing
- Any loss of income of carers or patients

We want you to tell us which program the decision makers should choose.

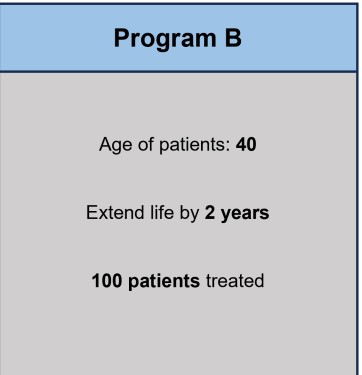

| **Program A** | **Program B** |
|---|---|
| Age of patients: **2** | Age of patients: **40** |
| Extend life by **2 years** | Extend life by **2 years** |
| **100 patients** treated | **100 patients** treated |

**Fig 2. Life Extension question example.**

questions both complemented the think-aloud in understanding participants' reasoning and also encouraged the participant to further reflect on their responses.

The interviews were conducted by three female interviewers aged between 27–53 years (AD, TPE and CB). AD had experience in conducting quantitative interviews with adults and received training at the start of the study. TPE and CB were experienced in conducting qualitative interviews. Regular debriefs following the interviews were conducted among the three interviewers. Discussions included reflecting on initial interviews, re-listening to interview recordings and evaluating interviewer prompts.

We conducted a two-stage pilot process to refine our questionnaire and interview approach. Initially, two pilot interviews were conducted with a convenience sample (known to the interviewers) to confirm interview prompts and processes; this data was not included in the analysis. Subsequently, the first six interviews were treated as a second pilot. The intention was to thoroughly assess these interviews and exclude the data if any issues arose. The pilot interviews were rewatched and discussed including a reflection on whether the prompts led to discussion which addressed the study questions. As no major changes were made to the interview prompts at this stage this pilot data is included in the main sample. Detailed information about the pilot and development of the survey prompts are described in the protocol paper [22].

Interviews took place between March and July 2023 and took 45 minutes on average to complete. The one-to-one interviews were conducted online using zoom with the interviewer entering responses to the survey via a shared screen. Participants had not met interviewers prior to the interview. Additional interviewer notes were made when necessary, during the interviews, including noting any technical problems and reactions from the participant to the survey questions.

At the start of the interview the interviewer explained how to participate in a think-aloud interview by demonstrating an example question about choosing a cat versus a dog as a pet. During the interview if the participants became quiet, they were encouraged to think-aloud and to explain why they had made their choice. The interviewers then completed the survey on behalf of the participants.

We acknowledge the potential bias from using interviewer-led mode. However, interviewer-led mode could help improve participant's comprehension of the survey tasks and facilitate responses. We had made several efforts to mitigate potential biases, first, using neutral interview prompts (e.g., "I would like you to talk me through your thought process – just saying whatever is coming into your head. There are no right or wrong answers or thoughts here") to ask survey questions, second, the survey began with an introduction video (S2 File) which talked through an example PTO question. In this video it was mentioned that participants could have views favoring young people or favoring adults.

Participants then answered seven PTO questions and three attitudinal questions and were invited to 'think-aloud' while they answered and to provide reasons behind their responses. They were also asked semi-structured questions to further probe their thinking and the reasons for their answers by answering feedback questions. The interviewers probed to explore differences in PTO responses between the different types of health gain. Where PTO responses appeared to give different preferences to attitudinal responses the interviewer asked for an explanation. If appropriate, the interviewer referred to other participants who had given apparently inconsistent views on these questions to ensure the participant did not feel their responses were being challenged. To explore the strength and robustness of the participants' views the interviewer presented contrasting opinions expressed by other participants (e.g., prioritizing children over adults or prioritizing adults over children) and asked how the current participant felt about these differing views. Interviews were video recorded and transcribed using automated intelligent verbatim transcription.

## 2.4 Qualitative research approach and data analysis

A thematic analysis was employed to identify the fundamental reasoning patterns in participants' responses. This approach allowed us to identify, analyze and report themes within the data. Our approach incorporated a framework analysis approach [30] as described in Figure 3 for data analysis. Three researchers (AD, TPE and CB) were involved in the data collection and analysis process. To ensure inter-coder reliability and data transparency we followed a detailed coding

**(1)Transcription**

Transcripts were uploaded to MAXQDA with the demographic information collected to facilitate the analysis.

**(2)Familiarization**

Three researchers (AD, TPE, CB) initially discussed 20 interviews to see whether additional information could be gathered and to become familiar with the data.

The researchers analysed the last three interviews to check whether saturation was achieved.

**(3)Coding**

Three researchers initially coded two transcripts independently to highlight broad themes in the interviews.

**(4)Developing the initial coding framework**

The three researchers met to discuss the initial coding. Similarities and differences were discussed and incorporated the changes to the existing codes.

**(5)Further Coding**

Further five transcripts were double coded, and AD, TPE and CB further discussed the list of codes and extra codes were added during these discussions.

**(6)Applying the coding framework**

One researcher (AD) applied the final coding framework to all the transcripts.

**(7)Thematic Analysis**

Further analysis was conducted to analyse the link between each theme. The most suitable quotations (1) which were concise, (2) which represented the codes well and (3) represented a range of different respondent's voices were selected by AD.

**Fig 3. Framework Analysis approach adapted from Gale et al. (2013).**

process. Initially, all three researchers independently coded two transcripts to highlight broad themes in the interviews. The researchers then met to discuss the initial coding, where similarities and differences were discussed, and changes were incorporated into the existing codes. A further five transcripts were double coded, and the researchers discussed the list of codes, adding extra codes during these discussions. One researcher (AD) then applied the final coding framework to all the transcripts.

Coding adopted both deductive (drawing upon the structure of the PTO survey and the study objectives) and inductive approaches. The final stages of the analysis included examining the coded data according to key demographic characteristics including gender, parental status, whether parents had children with health issues and employment status. We then linked these qualitative results to the quantitative findings.

## 3. Results

### 3.1 Participants

Recruitment was completed over a period of four months (27th March 2023 – 20th July 2023) after achieving thematic saturation. After 10 interviews were completed for each of the recruitment approaches (total 20), the researchers discussed the interviews to see whether new information could be gathered. Since new information was being generated, we continued conducting interviews. Our approach to determine saturation was based on thematic saturation and we considered saturation to be achieved when no new information relevant to the study objectives emerged. Once the researchers analyzed the last three interviews from the P-MIC sample, they decided that no new information relevant to study objectives was being generated and that we achieved thematic saturation. Data saturation was reached after the 38th interview.

A total of 41 participants were interviewed; 26 were recruited through CRNRstone and 15 from the P-MIC sampleframe. Of the 41 participants 56% were female and their mean age was 37 years (range: 16–86 years). Of the 26 participants recruited through the CRNRstone sample 11 were female. Of the 26, 7 were adolescents, 11 were adults with no children, 2 were parents with children with a health condition and 6 were parents with healthy children. Of the 15 participants recruited from the P-MIC sample, 12 were female and of the 15, 13 were parents with children with a health condition and 2 were parents with healthy children. Regarding employment status in the CRNRstone sample, 12 participants were employed/ self-employed, 9 were students, and 5 were retired. In the P-MIC sample, 12 were employed/ self-employed, 1 was on maternity leave and 2 were not working. The background characteristics of study participants are provided in supplementary material (S1 and S2 tables).

### 3.2 Qualitative results

Nine themes emerged from the codes, categorized within two main categories. These two categories cover the decision making process on PTO questions and attitudinal questions, respectively. An overview of the themes and sub themes is provided in Table 1. Additional quotes are provided in the supplementary material (Table S3).

**3.2.1 Category 1: Decision making process on PTO Questions.** We identified results relating to how participants arrived at their answers during the PTO questions. These results are described below including quotes taken from the transcripts. For context, the PTO trade of choice made by the participant is displayed after the quote and their chosen group is shown first.

**Interpretation of PTO Questions:** We identified three key themes relating to how participants interpret the size of gain and the impact beyond the individual presented in the trade off.

**Theme 1: Interpretation of Life Extension:** When answering the questions asking about two or five years of life extension, although the same amount of calendar time, the gain was interpreted by participants as bringing different experiences depending upon the age of the group. The participants interpreted the size of gain in life extension in three ways.

**Table 1. Thematic framework.**

| | Themes | Sub-themes |
|---|---|---|
| (1) Decision making process on PTO Questions | | |
| Interpretation of PTO questions | **1. Interpretation of Life Extension** | 1.1 Differences in the perceived experience of additional time by age |
| | | 1.2 Self-awareness of death |
| | | 1.3 Differences between an additional 2 and 5 years by age |
| | **2. Interpretation of quality of life** | 2.1 Ability to perform usual activities |
| | | 2.2 Ability to cope or adapt |
| | | 2.3 Ability to meet societal expectations or norms of their age socially |
| | | 2.4 Ability to understand the health condition |
| | **3. Impact beyond individual** | 3.1 Impact on family and parents |
| | | 3.2 Impact on society (in terms of earnings, tax) |
| Decision making criteria on PTO questions | **4. Decision making patterns observed** | 4.1 Drawing on own experiences |
| | | 4.2 Emotional response |
| | | 4.3 Life experiences |
| | | 4.4 Calculating the age of the group after treatment |
| | | 4.5 Calculating the most deserving based on largest proportional increase or through aiming to equalize lifetime opportunity |
| Challenges for participants in responding to PTO questions | **5. Some participants thinking about long term impact** | |
| | **6. Challenges in imagining a health scenario** | |
| | **7. Reluctance or discomfort making trade-offs** | |
| (2) Decision making process on Attitudinal Questions | | |
| Interpretation of attitudinal questions | **8. Perceived differences between attitudinal and PTO questions and underlying beliefs** | |
| Decision making criteria on attitudinal questions | **9. Decision making patterns observed for attitudinal questions** | 9.1 Priority based on equality of access<br>9.2 Priority based on largest gain<br>9.3 Priority based on other fairness criteria |

First, participants reflected on what they thought each age group could do and achieve with the additional time remaining, including why additional life expectancy is more, or less, valuable for particular ages.

For example, one participant interpreted the life extension of a 4-year-old as more important because *"all the life stages that you do get as children progress into teenagehood and, you know learning how to tie your shoelaces… And I think those sorts of memories are more valuable than extending the life of a 40-year-old"* [Male,27yrs: PTO 4 yrs/40 yrs]

Second, participants considered self-awareness of death. When choosing between older and younger groups to receive additional life years, four participants imagined the patient was aware of their death and considered how that knowledge would affect them, and how this might differ according to patient age. One participant interpreted the life extension by saying *"I think that if they knew that they were going to die at the end of the two years I would probably choose*

*the 40-year-old. But that's only because that would be a horrible thing for a 14-year-old to know"* [Female,37yrs: PTO 40 yrs/14 yrs]

Very few participants (4 participants) felt that very young children would never know or would never understand death and chose the older group to get the life extension treatment. For example, when comparing between patients aged 12 and 2 years old, one participant chose the 12-year-olds because *"a 2-year-old is not very aware of themselves and others…but a 12-year-old does understand the world and stuff"* [Male,39yrs: PTO 12 yrs/2 yrs]

Third, participants considered the differences between an additional 2 and 5 years by age. The life extension in the PTO consisted of two years and five years. Even though many (35 participants) participants did not change their responses between the two and five year life extension questions, six participants felt that these differences in duration would alter their preference.

One participant said that a two-year life extension would take a 12-year-old to 14 years and a five-year life extension would take them to 17 years. This participant explained the reason the two versus five years differs is that the five years takes the child to a different age group and the value of what they can experience with the additional time is greater.

*"If they were 12 years old and only had two years, they don't really care about going overseas. They want to see their friends and…mum and dad. Whereas the 17-year-old, you know they've been educated a bit about Bali or America or England"* [Male,65yrs: PTO 12 yrs/40 yrs]

**Theme 2: Interpretation of quality of life:**    This theme highlights how the health condition is perceived to impact on quality of life across different ages. Participants considered the ability to perform usual activities, ability to cope or adapt, ability to meet societal expectations or norms for age socially and the ability to understand the health condition.

Nine participants identified differences in a person's ability to work, go to school, do sports while having a health condition in different ages. For example, one participant interpreted that the quality of life of a 16-year-old would be more impacted because *"a 16-year-old is possibly involved in a lot of sport, very physically active, running around doing more…they can do that because their body is in physical condition that enables them to do that"* [Female,18yrs: PTO 16 yrs/40 yrs]. Another participant prioritized individuals aged 40 over those aged 8 by saying *"Because at 40 years of age, …those patients would be in the prime of their life as far as their work is concerned, and they probably need to be more fit and healthy as far as walking and moving around is concerned"* [Male,75yrs: PTO 40 yrs/8 yrs]. Out of the nine participants who reflected on the impact of the condition on usual activities, four were adolescents (16–18 years) one of whom explained that: *"people who are younger have to walk a lot more than someone who's 40 who can sit at their desk or do their job online if they need to…they don't have to drive or anything...So, someone who's 14 needs to get to school and do other things that they usually need to do"* [Female,17yrs: PTO 14 yrs/40 yrs]

Many participants (27 participants) used their perception of how the age group would cope with the condition as a way of prioritizing between groups. However, they had differing views on whether older or younger people would cope best. Nineteen participants chose the child age group because they interpreted the adults as better able to cope or adapt than children. For example, one participant thought 8-year-olds would not cope well with pain:

*"I'd probably choose the 8-year-old because….having pain would like be really hard to cope with being that young"* [Female,16yrs: PTO 8 yrs/55 yrs]

Conversely, eight participants perceived that children cope better than adults. For example, one participant considered a 4-year-old would cope better with low mood and anxiety than a 40-year-old because *"a 4-year-old, they've got someone around them 24/7. They've got a carer, or the parent would be able to distract them or give them tips and tricks…just give them an Icy pole…or go to the park… Whereas an adult you know, low mood and anxiety, stress…and left by themselves"* [Female,50yrs: PTO 40 yrs/4 yrs]

Very few parents with children with a health condition (3 participants) chose the adult age group considering that the child has sufficient support. For example, a parent suggested that a 55-year-old might cope less well in terms of low mood and anxiety because *"there is a much greater risk of lives lost and lives impacted if they are left untreated, whereas with a newborn because we've been through it, I know that there is support there and…earlier intervention with a one-month-old is a hell of a lot easier than with a full grown adult"* [Female,37yrs: PTO 55 yrs/one-month]

Participants interpreted the impact of the health condition in terms of how this will impact the patient socially, i.e., friend groups, self-confidence, self-esteem. For example, one participant said:

*"I think this is about….a 10-year-old being put through a symptom that could be resolved…. over a period of two years…So, whether it's low self-esteem, lacking confidence, you know not being popular at school…Those are very important, like formative years of a child, and I think they need to be protected"* [Male,37yrs: PTO 10 yrs/55 yrs]

Another participant chose the 8-year-old saying that for 8-year-olds *"being with like friends and playing and like going out…I feel like having…distress and low mood and anxiety would have a lot bigger impact"* [Female,16yrs: PTO 8 yrs/55 yrs]

Nine participants expressed that when the patients are not capable of understanding their health condition it would be more difficult for them to be in that condition therefore, they should be prioritized for treatment.

*"As a 2-year-old really doesn't know what pain is until he experiences and doesn't know what's happening to him. Whereas a 55-year-old would have experienced pain in the past and knowing that it's only going to last for two years and then he'll be back to normal health. Well, if I have to put up with it, I'll put up with it. But a 2-year-old just can't think along those lines. He doesn't know what's happening with him"* [Male,77yrs: PTO 2 yrs/55 yrs]

On the other hand, there were participants who expressed that understanding the health condition would make it worse therefore it would be beneficial to provide the treatment to those old enough to understand and remember.

*"I guess I'll choose the 40-year-old because I think they'll remember it. And I think that the one-month-old won't"* [Female,42yrs: PTO 40 yrs/one-month]

**Theme 3: Impact beyond individual:** Many participants (26 participants) considered the impact on family and society and how that may differ by age.

While making PTO choices participants didn't limit their thinking only to the patient but also thought about the impact on family. Participants made choices based on the **impact on family or children when their parent is ill** with a health condition.

*"Thinking like 40-year-olds, I could have...kids by then and I think it's important to be able to move if you have...kids or family"* [Female,16yrs: PTO 40 yrs/14 yrs]

*"I think if the patient (55 years) gets more pain, that means it may be affecting their well-being themselves, that means probably affecting the family"* [Female,50yrs: PTO 55 yrs/14 yrs]

Participants also thought about the **impact on family or children if the parent is dead** when responding to life extension questions.

*"I'll go program A (55-year-olds)…because they're parents they have families to support. More life would be more useful"* [Male,19yrs: PTO 55 yrs/14 yrs]

They also thought about how it would **impact the parents and family when the child is ill**. One participant preferred to prevent pain in a 2-year-old than an adult, explaining that "*the benefit is on to this child's side because I feel…it affects more people than the one*" [Male,39yrs: PTO 2 yrs/55 yrs]

Participants also considered what will happen to **parents or family if a child is given additional life years before their death** while making the tradeoff.

*"The reason I would choose program B* (4-year-olds) *would probably be just for the parents to give them more time with the kids because I feel…the kids wouldn't really appreciate that two years cause I guess life just goes on"* [Female,16yrs: PTO 4 yrs/40 yrs]

One participant focused only on the patient when it is a 55-year-old but focused on the parents when it's a 2-year-old: *"I'm thinking mainly the parents and friends, whereas the 55-year-old he could have retired…He's probably got all his family, might have grandchildren at that age. Parents of a 2-year-old…I guess if I was in that situation, I would want my child as long as I can to appreciate it"* [Male,77yrs: PTO 2 yrs/55 yrs]

How the impact on society in terms of productive labour and contribution to taxation differs by age was discussed by nine participants. For example, one participant chose the 40-year-old when considering preventing pain saying: *"I think I'll go with program A* (40-year-old)*…where those people are in the stage of their lives where they're working and they're being productive work-wise. They're productive in the community more than an 8-year-old"* [Male,75yrs: PTO 40 yrs/8 yrs]. Another participant chose 24-year-olds in life extending treatments because *"that's the entire tax paying life ahead of you"* [Female,43yrs: PTO 24 yrs/55 yrs]

**Decision making criteria on PTO questions:**    Of the 19 participants who completed the survey within the unforced-arm, four participants opted for the 'no preference' choice in all the PTO questions and one participant chose no preference in life extension questions. All these five participants were aged above 40 years.

*"I can't weigh up a 40-year-old's life versus a 10-year-old's life and say this person's more valuable than that person. I wouldn't want it done to me and I wouldn't want to do it to someone else"* [Female,42 yrs]

*"That is wicked…16-year-olds they're the future of our country…they've got parents who've loved them and nurtured them for 16 years, but then a 55-year-old is a parent of a 16-year-old. No, I don't want to have to be involved in making that sort of a ridiculous, unnecessary decision"* [Female,86yrs]

**Theme 4: Decision making patterns observed:**    This theme covers the patterns observed in participant responses who made trade-offs. Participants (1) drew on their own experiences, (2) used emotions, (3) considered life experiences, (4) calculated the age of the group after treatment and (5) calculated the most deserving based on largest proportional increase or through aiming to equalize lifetime opportunity.

Thirteen participants used their own experiences to inform their choices between children and adults and to explain why they made those choices. One participant used his experience as a Doctor in a hospital to make the trade off by saying *"In the past for 40 something years I worked in a Hospital (Australia), I graduated as a medical doctor long time ago and for me it doesn't matter what age, in front of me, it's a human"* [Male,77yrs: PTO 4 yrs/40 yrs]. Another participant used her own daughter's health condition to help make her decision; *"the reason why I think I'm finding it difficult is our daughter has a rare genetic condition and I know what happens when you are the one or the 25…and not getting that support"* [Female,38yrs: PTO 18 yrs/40 yrs]

Few participants (10 participants) had an emotional response to thinking about the suffering of a particular age group. These participants referred to their feelings towards the child or adolescent age group when describing how they made their decision. For example, one participant chose the one-month-old age group over adults because she "*would feel*

worse about it because they're babies. They've just been born into the world and they're suddenly experiencing a lot of pain" [Female,16yrs: PTO one-month/40 yrs]

Five participants tended to make PTO choices by considering the current age of the patient and what a person of that age would do normally, and hence the activities that would get impacted due to the health loss and choosing the group which would result in averting the greatest loss. One participant made the trade off by saying: *"You know, at the age of 20, you're just becoming an adult, you've got a lot of big things ahead of you. You might have a partner that you want to get married to. Whereas by the age of 40…you've had quite a while to do that already and most people by the age of 40 are married and have kids"* [Male,19yrs: PTO 20 yrs/40 yrs]

Participants also calculated the age of patients after treatment to help them make PTO trade-offs. One participant chose 40-year-olds because *"extending the life by two years, it's not a lot of time particularly to a 24-year-old…two years ago, they were only 22. In two years' time, they're only 26"* and she explained the reason was how much experience they would get by saying *"but then also looking at like what my parents were doing from 38 to 40, they were very much living their best life"* [Female,32yrs: PTO 40 yrs/24 yrs]

Very few participants (4 participants) based their PTO decisions on considerations of fairness. This included participants selecting the group that they calculated had the largest proportional increase in life expectancy.

*"Five years on a 2-year-old is like over 250%. Basically, we're adults…,55-5 years, 11% so it's not much"* [Male,39yrs: PTO 2 yrs/55 yrs]

Some participants (15 participants) adopted 'fair innings' considerations noting that adults (particularly the 55-year-olds) have already experienced their lives and have achieved things and the younger age group should be given the same opportunity. For example, one participant chose the 18-year-olds in favor of 55-year-olds because she feels *"like for 55 it's not fair…they've gotten like an extra like 30 something years to do things that they want to do…I just think it's not fair to give it to someone who's had more life, than someone who's had less"* [Female,16yrs: PTO 18 yrs/55 yrs]

**Challenges for participants in responding to PTO questions:** Three themes highlight the potential challenges faced by participants while responding to PTO questions.

**Theme 5: Some participants thinking about long term impact:** The PTO questions instructed participants to assume that the overall costs, carer's health and wellbeing and any loss of income of carers or patients are the same in both the programs and there is no long-term health impact. Although most of the participants adhered to these assumptions, there were participants who understood that they were asked not to consider long term effects but did think about how the health conditions might have a long-term impact in the future. Even with explicit instructions, participants found it difficult to detach from their understanding of real-world examples of long-term health impacts, illustrating the challenge in temporarily setting aside deeply ingrained health-related perceptions and experiences"

*"This bit where it says there are no long-term health consequences. But you can't be sure of that. You're sort of thinking…there will be some consequences that we don't know"* [Female,39yrs: PTO one-month/55yrs]

One participant chose 22-year-olds in favor of 55-year-olds reasoning that *"during that time your brain still hasn't fully developed, and so two years of mental health issues might end up having more long term damage"* [Female,18yrs: PTO 22 yrs/55 yrs]

**Theme 6: Challenges in imagining a health scenario:** Most of the participants were clear on the instructions and understood the questions. However, there were very few participants (5 participants) who sought the interviewer's confirmation regarding how severe the pain level is.

*"Do you know if that pain is like minor or severe?...is it kind of up to my own interpretation?"* [Male,27yrs: PTO 40 yrs/4 yrs]

Very few participants (4 participants) had difficulty imagining a one-month-old with a mental health condition. One participant said: *"I don't really see how you'd be able to diagnose a one-month-old with a mental health illness"* and because of this reason she chose the 40-year-old. [Female,16yrs: PTO 40 yrs/one-month]

**Theme 7: Reluctance or discomfort making trade-offs:**    Ten participants showed discomfort in undertaking PTO choices, but while reluctant to make a choice between children and adults, still made a choice. Most of these participants felt it was socially undesirable to make a trade-off.

"I don't know…I feel like a bad person" [Female,43yrs: PTO 24 yrs/55 yrs]

**3.2.2 Category 2: Decision making process on Attitudinal Questions.**  The survey included three attitudinal questions which are presented in [Figure 4](). Unlike the PTO questions these involved participant's beliefs on the Australian Healthcare system.

**Interpretation of attitudinal questions:**

**Theme 8: Perceived differences between attitudinal and PTO questions and underlying beliefs:**    This theme highlights how the participants interpreted the attitudinal questions and why their interpretation differed to the PTO questions.

Some participants traded off in PTO questions yet did not feel it was appropriate to prioritize either children or adults in attitudinal questions. One of the main reasons given for this was that they thought the age in attitudinal questions of below versus above 18 was too broad. One participant chose the 12-year-olds and 22-year-olds (compared to 55-year-olds) in

| Which of these statements best reflects your views about prioritising different types of health care? | Children should have some priority over adults. | Adults should have some priority over children. | People should have the same priority regardless of age |
|---|---|---|---|
| For medical care that improves quality of life temporarily (with no long-term effects) | | | |
| For medical care that extends life by a few years | | | |

If the Australian governments were willing to pay more for a treatment for children compared to adults which gave the identical health gain–what would you think?
○ This is fair because they are children
○ I'm not sure
○ This would be unfair

Which one of these statements best reflects your views about Medicare priorities?
○ Medicare should give priority to treating patients who will die young.
○ Medicare should give priority to treating patients who will get the largest amount of health benefit from treatment.
○ Medicare should give the same priority to treating all patients. Amount of health benefit and whether patients have had a short life is not relevant.
○ Medicare should base priority on a combination of treating patients who will get the largest amount of benefit and treating those who will die young.
○ Unsure
○ None of the above describes my views about Medicare prioritisation.

**Fig 4.  Attitudinal Questions in the Survey [22].**

the PTO questions but in the attitudinal questions they wanted to treat everyone equally. She explained this by saying that *"in terms of categorization when you're, labelling kids, it's only like up to about 18 or 20 versus when you're an adult, it's kind of 20s and above. So, it's a much larger population of people"* [Female,18yrs].

Another reason was that participants believed the reason the Australian healthcare system needs to make these tradeoffs is because there are not sufficient health resources. Because of this these participants argued there is a need to get more resources to treat both children and adults equally rather than prioritizing healthcare based on age. One participant said, *"the fight should be to get more resources rather than how to divide the limited resource you have"* [Female,43yrs]

**Decision making criteria on attitudinal questions:** The attitudinal questions involved age-based priority setting and participants drew their responses on their own understanding of Medicare. Medicare is the publicly funded healthcare scheme in Australia. All Australian citizens and permanent residents have access to fully covered healthcare in public hospitals, funded by Medicare, as well as state and federal contributions (Australian Government Department of Health and Aged Care, 2024).

**Theme 9: Decision making patterns observed for attitudinal questions:**

Participants based their decisions according to equality, largest gain or fairness in attitudinal questions.

Ten participants suggested that everyone must have equal access to Medicare regardless of age. One participant said *"my preference would be that we give a priority to all patients, regardless of whether or not their life is going to be short or not"* [Female,42yrs]

Two participants said Medicare should prioritize children and said *"it's fair because they're children and that childhood sets you up for the rest of your life"* [Female,18yrs]. One participant argued that Medicare should prioritize treatments which give positive outcomes, a participant said, *"It's just whether or not the treatments going to lead to an improvement in health or lots of years of extra life"* [Female,50yrs].

Participants also suggested Medicare should prioritize based on other fairness criteria such as ability to pay for health care. For example, one participant said Medicare *"should prioritize those who have the least assets and resources because they're the ones least able to afford it"* [Male,36yrs].

At the end of interviews, we discussed the opinions expressed by other participants in the study and related research to explore the participant's reaction to alternative viewpoints. Some of the participants (14 participants) understood and accepted the viewpoints by saying *"everyone has their own opinion, obviously, so if that's someone else's opinion, then sure"* [Female,45yrs]. There were very few (5 participants) who opposed other participants' viewpoints. When we provided a participant with an alternative view, she said: *"who's to say that I should get treatment before that fantastic little 16-year-old boy who belongs to my car club, who's just learned how to drive a car. But then who's to say he should get it before I should? No, he and I are two individual people…we should both be entitled to…healthcare either of us might need"* [Female,86yrs].

In the interviews it was evident that participants' views on healthcare prioritization were highly complex and divergent. While some common themes emerged, the perspectives and the reasons varied considerably, which highlights the complexity of this issue in healthcare prioritization.

## 4. Discussion

### 4.1 Summary

The findings from this qualitative study, which is a part of the wider PTO study [23], will enrich our understanding and interpretation of the quantitative components. The quantitative analysis of the mixed method PTO reported (1) children aged one month to 2 years old are given less weight than 40 or 55-year-olds in averting mental health problems, (2) all children (including one month old to 2 years) are given more weight than adults (aged 40 and 55) when considering

averting pain, (3) when the older age group is 55-year-olds (rather than 40 years old) participants are more likely to prioritize the younger age group and (4) the youngest participants are more likely to prioritize the younger age in PTO tradeoffs.

From the qualitative interviews, we found potential reasons behind the quantitative patterns reported by the wider PTO study [23]. We found that participants had difficulty conceptualizing a child aged one month old – 2 years old could be diagnosed with a mental health condition. Second, participants were emotional when thinking about a child being in pain and expressed not wanting to see them suffer. Similarly, Powell et al. [31] reported that adults believed being in pain or discomfort could be more challenging to a 10-year-old compared to adults. Third, the participants were more likely to draw upon 'fair innings' justification [6] for prioritizing adults when the adults were 55-year-olds, as they felt the 55-year-olds had already experienced life. Finally, it was evident that compared to adults, adolescents in our study considered the health of children or adolescents (mostly patients aged >10 years) as more important than that of adults and provided clear rationales; for example, ability to perform usual activities, to cope, and to meet societal expectations. An interesting finding was that younger people focused on the things that young people do (which are closer to their own experience), for example going to school, doing sports and travelling. They also thought that older people have already experienced life. Older participants focused more on the importance of adults' contribution to the family, work and society.

We also found some differences between parents and non-parents. For example, most of the parents with young children with a health condition prioritized the younger age. However, we also found that a few parents (with a child with a health condition) were more likely to prioritize 40 and 55-year-olds compared to children. The reasons for this relate to the importance of parents looking after their children and their beliefs that children have sufficient support and that it is more manageable for children.

The difference between attitudinal and PTO responses was mostly explained by our participants as being due to the broad categories used in the attitudinal questions that compare 'children' to 'adults', rather than ask about specific ages. The category 'children' includes some age groups participants may not wish to prioritize (i.e., the very young). Likewise, the category 'adults' includes some age groups they may wish to prioritize (i.e., young adults). However, it was also notable that the attitudinal questions encouraged participants to start thinking about broader funding issues of the Australian healthcare system.

One of the interesting findings in this study was observing how most participants reevaluate the size of the gain in the PTO question depending on the age of the group and selected the group with the largest gain. This occurred in both the quality of life and life expectancy questions. They considered what the patient would experience during their current age or considered what age they would be before and after treatments to understand what experiences they might miss if they do not get the treatment. This has not been reported in any previous PTO study. To our knowledge, this is the first study to document how participants actively reevaluate and contextualize health gains based on the age of the recipient of the treatment in PTO questions.

Our results also reported that 12% of the participants concluded that everyone should be treated equally by refusing to trade off and all these participants were above the age of 40. One of the reasons provided by the participants was that healthcare should not be given differently based on age and no matter how old the adult is everyone needs to be treated equally. Aidem [14] reported similar results where they reported that older participants believed that all patients need to be treated equally. Using information from focus group discussions, Kuder and Roeder [15] reported that the elderly should not be treated differently just because of age but if they were advised to at least choose one, they had some willingness to select the young person. Similarly, in our study we found some individuals (14 participants) who were reluctant to prioritize based on age.

Seven study participants chose the younger age group because they felt children/adolescents should experience more in life and they should get the opportunity to experience all the things that adults (40 or 55 years) have experienced. This is consistent with the findings from Schweda et al. [32], who reported some participants felt that extending a person's life was more important and legitimate if they are young because they could experience a more desirable life.

Even though, in the PTO task we explicitly stated that the health gain was the same for both age groups, a few participants did think about the long term benefits a child may get compared to adults. This aligns with scientific evidence supporting long-term benefits of health investments in early childhood [33].

Participants also made their trade off in PTO questions by examining how being in a health condition would affect their families and society as well- a finding also reported by Schweda et al. [32].

Dewilde et al. [34] recruited participants to value health states which enabled them to score the values on to a QALY scale. They explored why health state valuations differ by conducting think-aloud interviews and reported that participants considered that (1) children need to play and experience things in life, (2) adults need to take care of children, (3) children might be able to cope better and (4) children have difficulties in understanding poor health related quality of life (HRQoL). We find similar results and themes even though the study aims differed. We similarly found views on coping and understanding were important, but in our study, they could lead to either children being prioritized or adults depending on participant interpretation.

The PTO method allowed us to get a breadth of different participant perspectives which provides great insights into societal preferences. However, there were few participants (10 participants) who expressed discomfort in undertaking PTO choices which could potentially affect the validity of our results. Alternative methods, such as Discrete Choice Experiment (DCE) or Relative Social Willingness to Pay, do not present as stark a choice in relation to comparing patients of different ages, hence may generate less emotional response and discomfort for participants. These approaches were not chosen for this study in part because of the desire to focus on the issue of age weighting only and discuss this with participants [22]. It is unclear whether participants would experience similar discomfort completing these alternative approaches. However, the discomfort experienced by participants completing PTO studies may be mitigated to some extent in future studies by providing a no-preference option.

A recent systematic review [35] synthesized evidence on whether health state valuations differ between children vs adults. This review found on average adults were less willing to trade-off life years to avoid poor health states for children than for adults. Our study identified that respondents had multiple different considerations in terms of their interpretation of the impact of health-related quality of life states by age and the benefit of additional life years by age when answering PTO questions, suggesting complex thinking. Considerations varied by the age of the children with very young children sparking different considerations to older children. Many Time Trade-Off (TTO) studies adopt a single age for children of age 10, however, our findings suggest responses may differ if a different childhood age were to be used.

Our study focused on different ages in healthcare prioritization. However, it is worth reflecting on our findings in the context of the longstanding debate over the use of age weighting in the Global Burden of Disease (GBD) framework. When Disability-Adjusted Life Years (DALYs) were first introduced in 1990, they incorporated an age-weighting function that assigned greater weight to years lived between ages 9 and 54. This function began at zero at birth, peaked at approximately 1.52 around age 25, and declined to about 0.3 by age 100. The rationale was to reflect the social value of individuals at different stages of life—particularly the roles that young and middle-aged adults play in supporting the physical, emotional, and financial wellbeing of both younger and older generations [4]. As Murray and Acharya [36] explained: "The well-being of some age groups, we argue, is instrumental in making society flourish; therefore collectively we may be more concerned with improving health status for individuals in these age groups." Some participants in our study based their responses on the contribution to society of, particularly, young to middle aged adults. However, as discussed above, participants included many other considerations. In relation to equity, Murray and Acharya [36] noted that age weighting "does not discriminate between the lives of different individuals but simply differentiates periods of the life cycle for a cohort." However, our participants discussed the immediacy of prioritizing between different age patients and how it made them feel, and no participants considered the fairness of age-weighting if applied consistently to a cohort. Age weighting of DALYs was controversial from the outset, with critics (e.g., Anand and Hanson [37]) raising ethical objections to valuing human life differently based on age. In response to these concerns, the GBD removed age weighting from its methodology in 2010 [38].

## 4.2 Strengths and limitations

Some strengths of this study were that we included an adolescent sample to address the research gap identified through the review [16]. The review highlighted a lack of qualitative studies specifically examining adolescent's perspective on age-based healthcare prioritization. By including adolescents in our study, we have contributed to filling this gap in the literature and provided valuable insights into how younger individuals view healthcare prioritization across different age groups.

The study also had a large sample size and broad geographical representation. While the P-MIC sample focused on participants from Victoria, the CRNRSTONE sample included participants from New South Wales, Victoria, Queensland, Tasmania, South Australia, Western Australia and Australian Capital Territory. Furthermore, the study demonstrated breadth of coverage in terms of participant characteristics (56% females, 76% born in Australia, 17% adolescents).

Pilot interviews were carried out and the study was a combination of think-aloud and probing techniques. Think-aloud interviews helped us to better understand the participants' thought processes. Our study was conducted by three interviewers who could facilitate discussion and viewpoint diversity among the participants.

Nevertheless, this current study also has some limitations. The recruitment methods in our study may have introduced selection bias. Since the participants were recruited from a pool of participants willing to undertake qualitative research, this may potentially affect the generalizability of our findings. The study also includes unequal number of participants across categories (adolescents, parents, non-parents), which may potentially limit the diversity of perspectives captured. It is important to mention that this study was conducted involving participants from Australia, which might not be directly applicable to other social norms and values of other countries such as Asian, African countries. For example, existing literature indicates financial support, care support is provided by adult children for the wellbeing of older adults in China [39,40]. These differences in social values may impact healthcare prioritization decisions.

All interviewers were female, which may lead to potential response biases if some participants might have answered differently in relation prioritizing health care for children compared to adults. McNay [41] reported that gender has the most impact in determining the interactions between the interviewer and participant and the emotions of the participants. Male participants are also considered to develop their responses according to the gender of the interviewer [42]. Another potential limitation of this study is the lack of formal researcher reflexivity during the study, such as keeping a reflexive journal throughout. Informal discussions and reflections acknowledged that the three interviewers' backgrounds in health economics might have influenced their perspective on rational trade-offs in PTO choices. While the interviewers used standardized survey prompts described elsewhere [22] to explain that there are no right or wrong views, we acknowledge that this may not have fully mitigated potential biases occurring from their professional background. However, in the data interpretation approach, we ensured to consider a range of factors beyond economic optimizations. These included attention to participants' emotional responses, personal experiences and cultural influences.

In addition to this, there was a lack of depth of discussion relating to scrutinizing and challenging people's views. These discussions remained light touch given the sensitive nature of the topic, particularly for parents with children with health conditions. One of the study objectives was to explore whether the participants think the PTO questions were able to identify the relative weight they would give to improve child versus adult health. However, the survey prompts did not generate much discussion on this meta level concept and issues relating to this objective were not identified in the data. The study also aimed to understand the differences in responses to PTO vs attitudinal questions. Our findings suggests attitudinal questions encourage people to think about fairness of access and need for additional healthcare funding. However, the direct comparison was hindered by the broad age categories used in the attitudinal questions, given people's nonlinear preferences towards prioritizing across age groups.

## 4.3 Future research implications

The main aim of this study was to understand the reasoning behind the public's willingness (or not) to prioritize children's health gains over adult's health gains. More research is needed to understand whether replicating on different methods, e.g., Discrete Choice Experiment (DCE), Relative Social Willingness to Pay [10] or gain trade-off (GTO) would provide the same responses.

## 5. Conclusion

This study explored the views of Australian adolescents and adults on how they feel about valuing health gains differently based on age and for different types of health gains. Our study addresses four of the five objectives set out, providing valuable insights into the decision-making process of health-care prioritization.

First, we found that participants interpretation of PTO questions involved both cognitive thought processes and emotional considerations. These included reflection on personal experiences, analysis of fairness in life outcomes and processing of emotional responses to the age group presented. Second the study revealed that differences in PTO choices between prioritizing children or adults were largely driven by the way the participants interpret the impact of the HRQoL state or additional years of life. These interpretations were influenced by factors such as the size of the gain for each age group based on their ability to perform activities, their ability to cope or understand and anticipated life experiences. This highlights the importance of considering age-specific factors and the complexity of healthcare prioritization choices in healthcare decision making.

Third we identified inconsistencies between PTO and attitudinal questions were largely driven by participants consideration on fairness and access to healthcare. Additionally, when subjected to alternative viewpoints many demonstrated that they accept each other's opinions. These qualitative insights help to inform the development of different approaches in healthcare resource allocation, highlighting the importance of involving a diverse range of participants with varying views. Furthermore, the complex cognitive processes and varied interpretations revealed in the study highlight the need for more nuanced approaches to PTO methods.

While this study addresses four of its five objectives, it is important to note that one objective remains unexplored. Specifically, future research needs to address whether the participants believe that PTO questions effectively capture their relative weighting of child vs adult health improvements. This study did not generate sufficient discussion on this meta-level concept to allow us to draw conclusions.

## Supporting information

**S1 File. Coding tree.**
(PDF)

**S2 File. Introduction video transcript.**
(PDF)

**S1 Table. Characteristics of study participants.**
(DOCX)

**S2 Table. Differences in participants characteristics between recruitment methods.**
(DOCX)

**S3 Table. Additional quotes from participants.**
(DOCX)

**S4 Table. COREQ (COnsolidated criteria for REporting qualitative research) checklist.**
(DOCX)

**S3 File. Consent form.**
(PDF)

## Acknowledgments

QUOKKA-The following individuals are the Chief Investigators (CIs) of the QUOKKA project team: Nancy Devlin, Kim Dalziel, Brendan Mulhern, Gang Chen, Deborah Street, Julie Ratcliffe, Rosalie Viney, Richard Norman, Harriet Hiscock (Lead principal investigator: Nancy Devlin[1] [nancy.devlin@unimelb.edu.au])

The authors wish to acknowledge the helpful feedback and advice received on the conception and design of this study by QUOKKA's two principal advisory groups: our Decision Makers' Panel and our Consumer Advisory Group. We are also grateful to the Commonwealth Department of Health, Australia, for valuable guidance to the QUOKKA research program. We are also grateful to Lea Kevin-Tidis who provided administrative support. We would also like to thank colleagues, friends and family who gave up their time to complete pilot interviews and surveys.

## Author contributions

**Conceptualization:** Ashwini De Silva, Cate Bailey, Nancy Devlin, Richard Norman, Tessa Peasgood.

**Data curation:** Ashwini De Silva, Cate Bailey, Tessa Peasgood.

**Formal analysis:** Ashwini De Silva, Cate Bailey, Tianxin Pan, Tessa Peasgood.

**Funding acquisition:** Nancy Devlin, Richard Norman.

**Investigation:** Ashwini De Silva, Cate Bailey, Tessa Peasgood.

**Methodology:** Ashwini De Silva, Cate Bailey, Nancy Devlin, Richard Norman, Tianxin Pan, Tessa Peasgood.

**Project administration:** Ashwini De Silva, Cate Bailey, Tessa Peasgood.

**Resources:** Ashwini De Silva, Cate Bailey, Tessa Peasgood.

**Software:** Ashwini De Silva, Cate Bailey, Tessa Peasgood.

**Supervision:** Nancy Devlin, Richard Norman, Tianxin Pan, Tessa Peasgood.

**Validation:** Ashwini De Silva, Cate Bailey, Nancy Devlin, Richard Norman, Tianxin Pan, Tessa Peasgood.

**Visualization:** Ashwini De Silva, Cate Bailey, Nancy Devlin, Richard Norman, Tianxin Pan, Tessa Peasgood.

**Writing – original draft:** Ashwini De Silva.

**Writing – review & editing:** Ashwini De Silva, Cate Bailey, Nancy Devlin, Richard Norman, Tianxin Pan, Tessa Peasgood.

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
