## [Decision Letter · Decision Letter 0]

25 Mar 2025

PONE-D-25-04838Do you really want to see a 2-year-old suffer? Understanding people’s views on the relative value of health gains by agePLOS ONE?

Dear Dr. De Silva,

Thank you for submitting your manuscript to PLOS ONE. After careful consideration, we feel that it has merit but does not fully meet PLOS ONE’s publication criteria as it currently stands. Therefore, we invite you to submit a revised version of the manuscript that addresses the points raised during the review process.

We look forward to receiving your revised manuscript.

Kind regards,

Paolo Landa, Ph.D.

Academic Editor

PLOS ONE

Journal Requirements:

4. One of the noted authors is a group or consortium QUOKKA. In addition to naming the author group, please list the individual authors and affiliations within this group in the acknowledgments section of your manuscript. Please also indicate clearly a lead author for this group along with a contact email address.

5. We note that you have indicated that there are restrictions to data sharing for this study. For studies involving human research participant data or other sensitive data, we encourage authors to share de-identified or anonymized data. However, when data cannot be publicly shared for ethical reasons, we allow authors to make their data sets available upon request. For information on unacceptable data access restrictions, please see http://journals.plos.org/plosone/s/data-availability#loc-unacceptable-data-access-restrictions.

6. Please provide a complete Data Availability Statement in the submission form, ensuring you include all necessary access information or a reason for why you are unable to make your data freely accessible. If your research concerns only data provided within your submission, please write "All data are in the manuscript and/or supporting information files" as your Data Availability Statement.

7. One of the noted authors is a group or consortium QUOKKA. In addition to naming the author group, please list the individual authors and affiliations within this group in the acknowledgments section of your manuscript. Please also indicate clearly a lead author for this group along with a contact email address.

8. Please remove all personal information, ensure that the data shared are in accordance with participant consent, and re-upload a fully anonymized data set.

Additional Editor Comments :

Please address the reviewers comments provided

Reviewers' comments:

Reviewer's Responses to Questions

**Comments to the Author**

1. Is the manuscript technically sound, and do the data support the conclusions?

Reviewer #1: Yes

Reviewer #2: Partly

Reviewer #3: Yes

2. Has the statistical analysis been performed appropriately and rigorously?

Reviewer #1: N/A

Reviewer #2: Yes

Reviewer #3: Yes

3. Have the authors made all data underlying the findings in their manuscript fully available?

Reviewer #1: No

Reviewer #2: Yes

Reviewer #3: Yes

4. Is the manuscript presented in an intelligible fashion and written in standard English?

Reviewer #1: Yes

Reviewer #2: Yes

Reviewer #3: Yes

Reviewer #1: The study explores an important topic, providing insight into how people value health gains by age. However, several methodological and clarity issues need to be addressed:

• The title, "Do you really want to see a 2-year-old suffer?", may be misleading. The study examines preferences for health gains across a broader age range (1 month to 24 years) rather than focusing solely on very young children.

• The study relies on interviews conducted while participants completed questionnaires, which may introduce bias. Participants might feel pressured to prioritize children or refrain from explicitly stating they do not prioritize based on age. Clarification is needed on how potential biases were addressed.

• Recruitment method is not described in detail. If participants were not randomly selected, selection bias could impact the generalizability of findings (Lines 221-222).

• Definition of “parents of children with a health condition” (Line 132): The restriction to children aged 19 and above is unclear. Why exclude parents of younger children with health conditions?

• Recruitment details (Lines 145-147): The description does not mention parents of healthy children, yet they were included in the study. Clarification is needed.

• Representativeness of the sample (Lines 148-150): The sample draws from the P-MIC study, which included parents of children aged 2-18 at a single hospital. This limits generalizability of the population.

Grammar and Clarity Issues

• Line 248: "participants considered self-awareness of death When choosing between..."

• Line 273: "Nine participants identified differences in a person’s their ability to work."

• Lines 501-502: "explained by our participants as being due the broad categories used in the attitudinal questions."

• Line 540: TTO typo

Reviewer #2: This is a well-conducted research article, but I have the following questions and suggestions for improvement:

1.Rationale for Sample Size:

The quantitative section (PTO questionnaire) should include a more detailed justification for the sample size calculation, such as power analysis parameters (e.g., α value, effect size, and target power).

Clarify the data saturation criteria for qualitative interviews, particularly for small sample subgroups (e.g., adolescents, n = 7; parents of sick children). Specify whether saturation was determined by theme repetition rate or the occurrence of negative cases.

2.Justification for Age Selection:

Explain in the methodology or discussion why the study focuses on 2-year-olds rather than other age groups (e.g., 3 or 4 years old).

If the selection is based on policy guidelines (e.g., Australian Pediatric Guidelines' age thresholds), please cite the relevant sources.

3.Results Optimization:

Chart Clarity: Figures 1 and 2 (schematic diagram of case issues and thematic framework) appear vague, making the text difficult to read. Consider improving clarity and resolution for better comprehension.

4.Negative Case Reporting:

The results primarily support the "children first" argument. To ensure a balanced discussion, include citations of perspectives opposing this principle (e.g., views prioritizing adult productivity). This will help mitigate potential selection bias.

5.Discussion on Cultural Generalizability:

Consider discussing whether findings from the Australian sample apply to collectivist cultures, such as East Asian societies, which may prioritize elder care. This should be acknowledged as a potential limitation.

6.Ethics and Data Transparency:

While the study states that "verbatim transcripts of interviews are ethically restricted from publication," consider making anonymized subject codebooks publicly available to enhance methodological transparency and reproducibility.

Reviewer #3: Thank you for the opportunity to review this manuscript, which qualitatively examines societal perspectives on age-related prioritization in healthcare resource allocation. The authors have approached the research question thoroughly and I enjoyed reading this manuscript.

Abstract:

The abstract is clear and well-structured. However, it would enhance clarity if the authors explicitly mentioned how these qualitative insights could inform resource allocation decisions.

Introduction:

The introduction effectively situates the study but could benefit from deeper theoretical grounding. Specifically, it would be valuable to explicitly reference ethical theories (e.g., "fair innings" theory, capability approach, or distributive justice theories) to contextualize age-based prioritization debates.

Methods:

- Recruitment (Lines 146-147):

It is noted that two recruitment methods were used (CRNRSTONE and QUOKKA’s P-MIC sample). Please explicitly discuss potential differences in socioeconomic or demographic characteristics between these recruitment strategies and their implications for the study. Consider adding a detailed table comparing demographic characteristics between groups.

Survey Design

Line 156 references a protocol, which is referenced elsewhere in the paper as established evidence that adolescents coped well with interviews. This does not seem correct, can you clarify? What were the age groups of the pilot interviews?

Lines 170-171. Figures 1 and 2 in the appendix are not legible.

Data collection: The authors describe that there were two pilots and iterations were made. What were the changes made?

The authors do not describe any reflexive approaches, they also do not describe the potential biases they may bring into this research as a result of their positionality. This is later described in the limitations section, but more information is still needed on positionality and reflexive approaches taken to mitigate potential bias.

Data analysis: The data analysis section (lines 208-) are sparse and do not describe how themes were chosen. “The final stages included analyzing the data according to demographic characteristics of the

participants and linking the qualitative results to the quantitative findings” what demographic characteristics?

There are no details on how data were coded into themes under the analysis section, by whom, and whether disagreements or reliability in coding were considered. Provide your resulting coding framework as per your method.

Results

Eleven themes are alot for qualitative research. Some of these themes seem very minor. For example, Theme 10 appears to only be supported by a few participants. This is not a major recurrent theme and should be moved to a minor section. Throughout, it is sometimes unclear whether themes are well supported by the data, or if this is a minor divergent theme. The distinction between major and minor, and divergent cases should be made. When divergent cases are included, ensure that they directly relate to your research question. This is also related to aim 5

- Saturation (Lines 219-222):

Clarify precisely when thematic saturation was reached. Current wording implies saturation occurred around the 39th-41st interview. A concise statement indicating exactly at which interview thematic saturation occurred would enhance methodological rigor. Further (related to above point) because there were 11 themes which is more than the average qualitative paper, when was saturation for each reached? Did some themes reach saturation more quickly than others?

- PTO methods:

Some participants expressed discomfort or uncertainty when responding to PTO questions. The manuscript should critically address whether PTO remains suitable given participants' discomfort and the emotional bias introduced. Could the authors discuss alternative methods (e.g., Discrete Choice Experiments or Relative Social Willingness-to-Pay) and justify their continued use of PTO despite these limitations?

Results:

- Adolescent vs parent opinions:

The manuscript clearly highlights differences between adolescents’ and parents’ prioritizations. It would strengthen the analysis to explicitly link these differences to the theoretical or empirical literature on intergenerational equity.

- Interpretation of PTO questions:

Themes identified are insightful but primarily descriptive. The analysis would benefit from deeper interpretation. For example, how do these themes inform our broader understanding of resource allocation, especially concerning fairness, equity, and efficiency?

- Minor Errors:

Line 540 incorrectly refers to TTO instead of PTO, which is confusing and should be corrected.

Discussion:

While QALYs do not typically incorporate explicit age weights, discussing the ethical debates around age weighting in DALYs (as done historically) would be relevant here. Explicitly mention pros (e.g., potential societal consensus on fairness) and cons (e.g., equity concerns, comparability) associated with explicit age weighting as per Murray et al. (2012) and Vos et al. (2012).

Provide greater integration of qualitative findings with the quantitative results from your companion quantitative PTO study. Specifically, explain clearly how qualitative data enrich or challenge the quantitative conclusions.

- Limitations:

Explicitly discuss how the single-gender (female) of interviewers might have influenced participant responses, particularly given the sensitive nature of the topic.

Conclusion

The conclusion does not cover the 5 aims discussed in the introduction. Include a brief overview of the ways in which the five aims are concluded. There are bits a pieces throughout the manuscript, but there is sometimes an unclear link between the aim and the results/discussion. When I read the conclusion, I am not sure that I walk away with a clear understanding of the aims.

Minor issues:

- Reference [29] is an unpublished manuscript currently unavailable online. Clarify its accessibility or update with available published references if possible.

- Consistently clarify the terminology regarding age categories to avoid confusion between “children,” “adolescents,” and “young people.”

**Do you want your identity to be public for this peer review?** For information about this choice, including consent withdrawal, please see our Privacy Policy

Reviewer #1: No

Reviewer #2: No

Reviewer #3: No

---

## [Author Response · Author response to Decision Letter 1]

15 May 2025

Addressing Reviewer Comments

*All changes made are shown in italics and underlined.

Dear Dr Paolo Landa,

I sincerely appreciate your thoughtful comments and feedback on our manuscript, your insights have been invaluable in strengthening our work, and we are grateful for the time and effort you have dedicated to reviewing it.

Reviewer #1

The study explores an important topic, providing insight into how people value health gains by age. However, several methodological and clarity issues need to be addressed.

Thank you for your thoughtful feedback and time in reviewing the manuscript.

1) The title, "Do you really want to see a 2-year-old suffer?", may be misleading. The study examines preferences for health gains across a broader age range (1 month to 24 years) rather than focusing solely on very young children.

Response: Thank you for raising this point. The current title includes a quote from the respondents. The current title highlights a powerful emotional response from participants, which is a finding of this study, however we agree it could be misleading therefore the title is now changed; “A qualitative study to understand people’s views on the relative value of health gains for children and young people compared to adults”

2) The study relies on interviews conducted while participants completed questionnaires, which may introduce bias. Participants might feel pressured to prioritize children or refrain from explicitly stating they do not prioritize based on age. Clarification is needed on how potential biases were addressed.

Response: Thank you for highlighting this methodological issue. We had considered the potential biases and several efforts have been made during the interview process. First, interview prompts were used to ask survey questions as neutrally as possible. Second, we made an introduction video and presented it at the beginning of the survey, where we mentioned that participants could have views favouring either young people or adults. Third, at the end of the interview, interviewers discussed with participants about alternative opinions of other participants (this was already mentioned in the main text line 246-253).

We have now included this in the manuscript under the section Data Collection.

“At the start of the interview the interviewer explained how to participate in a think-aloud interview by demonstrating an example question. During the interview if the participants became quiet, they were encouraged to think-aloud and to explain why they had made their choice. The interviewers then completed the survey on behalf of the participants. We acknowledge the potential bias from using interviewer-led mode. However, interviewer-led mode could help improve participant’s comprehension of the survey tasks and facilitate responses. We had made several efforts to mitigate potential biases, first, using neutral interview prompts (e.g., “I would like you to talk me through your thought process – just saying whatever is coming into your head. There are no right or wrong answers or thoughts here”) to ask survey questions, second, the survey began with an introduction video which talked through an example PTO question. In this video it was mentioned that participants could have views favoring young people or favoring adults.”

3) Recruitment method is not described in detail. If participants were not randomly selected, selection bias could impact the generalizability of findings (Lines 221-222).

Response: Thank you for raising this point. In the manuscript we have mentioned the two recruitment methods in (line 181-189). We employed two recruitment methods: 1) through a commercial company, CRNRSTONE and 2) participants in the QUOKKA Paediatric Multi-Instrument Comparison Study Protocol (P-MIC) study (parents who had previously consented for future research by the Royal Children’s Hospital)

We have now included more information regarding the recruitments, and it now reads as follows:

“Participants were recruited through two mechanisms to ensure inclusion of a breadth of perspectives. The first sample of participants mainly focused on adolescents and adults without children and was recruited through a commercial company, CRNRSTONE. CRNRSTONE invited participants on their panel who had expressed an interest in taking part in qualitative research. The second sample focused on parents of children who had experienced a health issue at some point and was recruited from participants in the QUOKKA Paediatric Multi-Instrument Comparison Study Protocol (P-MIC) study [28] who had previously consented to be contacted for future research. Parents of healthy children were included in both the samples.”

We have now included this as a potential limitation of the study.

“Nevertheless, this current study also has some limitations. The recruitment methods in our study may have introduced selection bias. Since the participants were recruited from a pool of participants willing to undertake qualitative research, this may potentially affect the generalizability of our findings.”

4) Definition of “parents of children with a health condition” (Line 132): The restriction to children aged 19 and above is unclear. Why exclude parents of younger children with health conditions?

Response: Thank you for highlighting this issue. The age mentioned within brackets are the age of the parents and not the age of children. We have amended this and the text now reads as,

“Therefore, our recruitment ensured coverage of non-parents, parents of healthy children, and parents of children with a health condition across different age groups (aged from 19 years and above).”

5) Recruitment details (Lines 145-147): The description does not mention parents of healthy children, yet they were included in the study. Clarification is needed.

Response: Thank you for raising this. The sample recruited through CRNRSTONE and QUOKKA P-MIC samples did include parents of healthy children. However, in line 182-189 we have mentioned the main focus of the sample. We have made these sentences clearer and it now reads,

“The first sample of participants mainly focused on adolescents and adults without children and was recruited through a commercial company, CRNRSTONE. CRNRSTONE invited participants on their panel who had expressed an interest in taking part in qualitative research. The second sample focused on parents of children who had experienced a health issue at some point and was recruited from participants in the QUOKKA Paediatric Multi-Instrument Comparison Study Protocol (P-MIC) study [28] who had previously consented to be contacted for future research. Parents of healthy children were included in both the samples.”

6) Representativeness of the sample (Lines 148-150): The sample draws from the P-MIC study, which included parents of children aged 2-18 at at a single hospital. This limits generalizability of the population.

Response: Thank you for raising this. To clarify on this the first recruitment through CRNRSTONE does include parents (with children aged >18), non-parents). Therefore, we believe the two recruitment methods allowed a wide range of population characteristics as mentioned in S1 table.

The Royal Children's Hospital is the largest comprehensive children’s hospital in Victoria. We have now included this in the manuscript under recruitment and participants.

“The P-MIC study included 1000 parents or caregivers of Australian children and adolescents aged 2-18 who attended the Royal Children’s Hospital (RCH). The RCH is the largest comprehensive children’s hospital in Victoria, Australia. It covers a broad range of health conditions [28].”

7) Grammar and Clarity Issues

Response: Thank you for raising this. These issues have been amended.

• Line 248: "participants considered self-awareness of death When choosing between..."

“Second, participants considered self-awareness of death. When choosing between older and younger groups to receive additional life years….”

• Line 273: "Nine participants identified differences in a person’s their ability to work."

“Nine participants identified differences in a person’s their ability to work, go to school, do sports while having a health condition in different ages.”

• Lines 501-502: "explained by our participants as being due the broad categories used in the attitudinal questions."

“The difference between attitudinal and PTO responses was mostly explained by our participants as being due to the broad categories used in the attitudinal questions……”

• Line 540: TTO typo- This is not a typo, what we wanted to show was differences between methods TTO and PTO. We have mentioned it in full form to avoid confusion.

““Many Time Trade-Off (TTO) studies adopt a single age for children of age 10, however, our findings suggest responses may differ if a different childhood age were to be used.”

Reviewer #2

This is a well-conducted research article, but I have the following questions and suggestions for improvement.

Thank you for taking your time to review the manuscript and thank you for your feedback.

1) Rationale for Sample Size:

The quantitative section (PTO questionnaire) should include a more detailed justification for the sample size calculation, such as power analysis parameters (e.g., α value, effect size, and target power).

Response: We appreciate your suggestion to elaborate on the sample size. The quantitative study (protocol and results) are written up separately and this paper is just the sister qualitative paper with a separate recruitment approach. The protocol paper is already published [22].

We have mentioned that we expected saturation to be achieved at 40 samples on the basis of Ritchie et al. [20] who suggested fewer than 50 will be adequate for individual interviews. We have now included more references to support the rationale behind 40 samples. The text now reads:

“We anticipated saturation would be reached at a sample of 40 interviews on the basis of Ritchie et al. [25] who suggested that studies involving a very diverse population might require an increased sample size, but a sample of fewer than 50 will be adequate for individual interviews. This is further supported by Hennink et al [26] who suggested saturation would achieve between 16-24 interviews and Guest et al [27] who found saturation occurred within 12 interviews.”

Clarify the data saturation criteria for qualitative interviews, particularly for small sample subgroups (e.g., adolescents, n = 7; parents of sick children). Specify whether saturation was determined by theme repetition rate or the occurrence of negative cases.

Response: Thank you for raising this. The sampling was designed to ensure a breadth of perspectives and viewpoint diversity, we have therefore removed the use of the term sub-group which was deceptive as saturation was determined based on the overall sample. We have amended this section.

“Participants were recruited through two mechanisms to ensure inclusion of a breadth of perspectives.”

“Our approach to determine saturation was based on thematic saturation and we considered saturation to be achieved when no new information relevant to the study aims emerged. Once the researchers analyzed the last three interviews from the P-MIC sample, they decided that no new information relevant to study objectives was being generated and that we achieved thematic saturation. Data saturation was reached after the 38th interview.”

2) Justification for Age Selection:

Explain in the methodology or discussion why the study focuses on 2-year-olds rather than other age groups (e.g., 3 or 4 years old). If the selection is based on policy guidelines (e.g., Australian Pediatric Guidelines' age thresholds), please cite the relevant sources.

Response: A number of the included quotes are based on comparisons to 2-year-olds, however, a range of different ages are included in the study. The PTO questions ask about children and young people ages 0,2,4,6,8,10,14,16,18,20,22,24 compared with adults aged 40 or 55. Each participant was asked questions about one children/youth age (randomly assigned) compared to one adult age (randomly assigned). The section in the discussion (line 535-547) links to the quantitative results of this study. The information regarding the age categories are mentioned in line 200-202 (“Participants were asked to make choices between pairs of interventions, one impacting an adult group (either 40 or 55 years old) and the other younger group consisting of 13 age categories (one month, even number of years between 1 year and 24 years old”).

3) Results Optimization:

Chart Clarity: Figures 1 and 2 (schematic diagram of case issues and thematic framework) appear vague, making the text difficult to read. Consider improving clarity and resolution for better comprehension.

Response: Thank you for raising this. We apologize for the lack of clarity in figure 1 and figure 2. We have now amended this.

4) Negative Case Reporting:

The results primarily support the "children first" argument. To ensure a balanced discussion, include citations of perspectives opposing this principle (e.g., views prioritizing adult productivity). This will help mitigate potential selection bias.

Response: Thank you for your suggestion. In the results section, we have mentioned quotes from participants supporting both views.

• In the quotes mentioned under the PTO trade-offs, 20 are from decisions in which the younger group was favoured and 10 were from when the older group was favoured.

• The reason behind choosing these quotes was to indicate the range and breadth of views and not to focus on the number holding any particular views. However, the quantitative sister paper does report the number.

• We have also added a section to the end of the results section.

“In the interviews it was evident that participants’ views on healthcare prioritization were highly complex and divergent. While some common themes emerged, the perspectives and the reasons varied considerably, which highlights the complexity of this issue in healthcare prioritization.”

• We also included an additional quote on productivity.

“Nine participants identified differences in a person’s their ability to work, go to school, do sports while having a health condition in different ages. For example, one participant interpreted that the quality of life of a 16-year-old would be more impacted because “a 16-year-old is possibly involved in a lot of sport, very physically active, running around doing more…they can do that because their body is in physical condition that enables them to do that” [Female,18yrs: PTO 16 yrs/40 yrs]. Another participant chose the 40-year-olds over the 8-year-olds by saying “Because at 40 years of age, …those patients would be in the prime of their life as far as their work is concerned, and they probably need to be more fit and healthy as far as walking and moving around is concerned” [Male,75yrs: PTO 40 yrs/8 yrs].”

5) Discussion on Cultural Generalizability:

Consider discussing whether findings from the Australian sample apply to collectivist cultures, such as East Asian societies, which may prioritize elder care. This should be acknowledged as a potential limitation.

Response: Thank you for the suggestion. We have included this as a potential limitation.

“Nevertheless, this current study also has some limitations…. It is important to mention that this study was conducted involving participants from Australia, which might not be directly applicable to other social norms and values of other countries such as Asian, African countries. For example, existing literature indicates financial support, care support is provided by adult children for the wellbeing of older adults in China [38,39]. These differences in social values may impact healthcare prioritization decisions.”

6) Ethics and Data Transparency:

While the study states that "verbatim transcripts of interviews are ethically restricted from publication," consider making anonymized subject codebooks publicly available to enhance methodological transparency and reproducibility.

Response: Thank you for your suggestion regarding data transparency. We appreciate the importance of methodological transparency and reproducibility in research. However, according to the ethics appr

---

## [Decision Letter · Decision Letter 1]

2 Sep 2025

Dear Dr. De Silva,

Thank you for submitting your manuscript to PLOS ONE. After careful consideration, we feel that it has merit but does not fully meet PLOS ONE’s publication criteria as it currently stands. Therefore, we invite you to submit a revised version of the manuscript that addresses the points raised during the review process.

**ACADEMIC EDITOR: The reviewers found many elements to modify in order to improve your manuscript.**

We look forward to receiving your revised manuscript.

Kind regards,

Paolo Landa, Ph.D.

Academic Editor

PLOS ONE

Journal Requirements:

Reviewers' comments:

Reviewer's Responses to Questions

**Comments to the Author**

Reviewer #4: (No Response)

Reviewer #5: All comments have been addressed

Reviewer #6: All comments have been addressed

Reviewer #7: All comments have been addressed

Reviewer #8: (No Response)

Reviewer #9: All comments have been addressed

Reviewer #10: (No Response)

Reviewer #11: All comments have been addressed

Reviewer #12: (No Response)

Reviewer #13: All comments have been addressed

Reviewer #14: (No Response)

2. Is the manuscript technically sound, and do the data support the conclusions?

Reviewer #4: Yes

Reviewer #5: Yes

Reviewer #6: Yes

Reviewer #7: Yes

Reviewer #8: Yes

Reviewer #9: Yes

Reviewer #10: Partly

Reviewer #11: Yes

Reviewer #12: Yes

Reviewer #13: Yes

Reviewer #14: Yes

3. Has the statistical analysis been performed appropriately and rigorously?

Reviewer #4: Yes

Reviewer #5: Yes

Reviewer #6: Yes

Reviewer #7: Yes

Reviewer #8: Yes

Reviewer #9: N/A

Reviewer #10: Yes

Reviewer #11: Yes

Reviewer #12: Yes

Reviewer #13: N/A

Reviewer #14: Yes

4. Have the authors made all data underlying the findings in their manuscript fully available?

Reviewer #4: Yes

Reviewer #5: Yes

Reviewer #6: Yes

Reviewer #7: Yes

Reviewer #8: Yes

Reviewer #9: Yes

Reviewer #10: Yes

Reviewer #11: Yes

Reviewer #12: No

Reviewer #13: No

Reviewer #14: Yes

5. Is the manuscript presented in an intelligible fashion and written in standard English?

Reviewer #4: Yes

Reviewer #5: Yes

Reviewer #6: Yes

Reviewer #7: Yes

Reviewer #8: Yes

Reviewer #9: Yes

Reviewer #10: Yes

Reviewer #11: Yes

Reviewer #12: Yes

Reviewer #13: Yes

Reviewer #14: Yes

Reviewer #4: Additional Comments:

Dear author,

I understand the importance of evaluating the personal rules involved in choices regarding investments in health programs by age.

In your article, I missed the addition of scientific studies on health outcomes related to the age of patients which should be also included to help decision-making. See, for example, articles on investments in health in early childhood. I suggest that you add some references to articles in this sense and that you mention the importance of such a source of decision-making in health.

Reviewer #5: Authors addressed all comments raised by the reviewer in beautiful manner. Now this manuscript is ready for publication.

Reviewer #6: Title: The title precisely reflects the subject of the study; however, it is academically advantageous to consider substituting the term "people" with "public."

“A qualitative study to understand public views on the relative value of health gains for children and young people compared to adults”

Abstract:

• Methods: Concise overview. Nonetheless, I suggest the inclusion of the following statement at the last line of the methods used paragraph: “Thematic analysis was employed to identify fundamental reasoning patterns.”

Introduction:

The introduction is thorough and based in robust research. However, Structural enhancements are necessary to improve the clarity and scholarly tone, such as:

• Lines (108 –128), (129-142), The paragraphs are too dense and long, consider break them into clear paragraphs.

• Line 113, those who have not had their fair innings needs to be prioritized. Change needs to “need”

• Line 120, consider changing >5 years to “older than 5 years” for more academic tone.

Methods: Demonstrate methodological transparency, provide a strong rationale for recruitment, and offer clear justification for the selections. Nevertheless, it requires slight modifications for improved clarity, formatting, and organization.

• Recruitment and Participants: Lines (164-179) and (180-192) consist of extensive paragraphs; please divide them into shorter paragraphs.

• Survey Design: The paragraph is extensive, consisting of lines 198-210. It is desirable to divide it into two paragraphs.

• Data Collection: The paragraphs lines (218-227) and (233-253) are extensive; therefore, it is necessary to divide them into separate paragraphs.

Results:

• Participants section: It is well written; however, please divide it into paragraphs for enhanced readability.

• Qualitative Results: it is well organized and comprehensive, but consider this:

a) rewrite the sentence “Another participant chose the 40-year-olds over the 8-year-olds” in the line 333, for more academic tone “participants aged 40 over those aged 8”

b) line 337 change the statement “people who are younger have to walk a lot more than someone who's 40” to “Younger participants were viewed as more reliant on physical mobility, compared to those who are 40 years old”

c) Theme 5: Provide a short analytical statement elucidating the reason for this and why it happened for instance, “Despite the guidance, participants struggled to disengage from real-world connections to chronic illness, illustrating the psychological challenge of ‘bracketing’ prolonged considerations.”

Discussion: This discussion is substantial and effective, but it can be enhanced through refinement in:

• lines (535-554) and (637-662), are lengthy, please divide them into paragraphs.

• Line 566-571 showing your novel contribution to the literature and the readers need to see this, so please add a statement that clarify this in the line 571 such as “To our knowledge, this is the first study to document………”

• Line 542, change the informal statements please “unable to imagine” to “had difficulty conceptualizing”

Conclusion: It offers clear alignment with study objectives, acknowledges cognitive and emotional complexity, and is relevant to policy and methodology. However, please consider some amendments such as:

• Line 674, These interpretations were reasoned by factors such as… change to These interpretations were influenced by factors such as… (for an elevated research tone).

• Replace informal language with academic ones such as line 671 “drawing on emotional reaction” to “triggering emotional response”, and line 676 “what they might experience if they live” to “anticipated life experiences”

• Add word “Additionally” before (4) When subjected to alternative viewpoints… (To enhance the coherence between sentences for improved readability).

• The conclusion indicates that four out of five aims were addressed; however, it does not provide a brief explanation for the aim that was not addressed. A brief mention would enhance transparency.

• The final sentence suggests practical relevance; however, the conclusion would be enhanced by adding a strong closing statement regarding the implications of the findings for healthcare policymakers, PTO methodology, or valuation research. Please consider adding this.

Reviewer #7: Thank you for the invitation to review. The authors have appropriately revised their manuscript in accordance with the previous reviewers' requirements. I believe the manuscript now meets the standards for publication in PLOS ONE, and I recommend its acceptance.

Reviewer #8: I have some few observations on the study title which to be address by the authors to facilitate reading and better understanding by the authors

A qualitative study to understand some selected individual's views on the relative value of health gains for children and young people in..... (here the name of the exact place where the study took place must be mentioned.) in Australia compared to Adults .....(which year was the study conducted )?

Methods

Under line 181 which says that adolescents and adults without children -Can the authors describe and elaborate further on what they meant by without children whether biological or adopted children or had children but are not staying with them (empty net syndrome )

Pre-testing of questionnaire,

were the questionnaire pre-tested and where were they pre-tested (pretesting of questionnaire cant take place in the study area) .

In line 631 there is an incomplete statement

some strengths of the study were that we included an adolescent sample to address the research gap identified review by...this part of the statement must be concluded with purpose and meaning.

Conclusion.

Since the study was not conducted in the entire Australia the Authors cant conclude that the study explored the views of Australians but a statement can be made to cover the exact geographical location where the study took place.

check line 630 where you talked about addressing four of the five aims set out,can the authors explain why the 5th aim was not address in the study or why it is not mention in the write-up

Reviewer #9: I consider this to be an interesting article, and one that warrants further work on the subject. I believe that the authors have taken the reviewers' corrections seriously, which has helped them to improve their manuscript.

Reviewer #10: Thank you for submitting this interesting manuscript entitled “Spatial epidemiology of dengue in northeastern Thailand: A Bayesian approach.” The study addresses an important public health issue and employs Bayesian spatial modeling, which is an appropriate and modern approach for investigating dengue distribution. The manuscript is promising, but several revisions are needed before it is suitable for publication.

1. Study design and data sources

- The use of surveillance data is appropriate, but please elaborate more on the limitations such as underreporting, case definition, and ecological fallacy.

- Clarify whether population denominators and case counts were adjusted for potential reporting inconsistencies across districts.

2. Bayesian modeling and statistical details

- The CAR/BYM approach is suitable, but details on prior distributions, convergence diagnostics (e.g., trace plots, R-hat, ESS), and sensitivity analysis should be added.

- Explain more clearly how model fit was assessed and whether alternative specifications were tested.

3. Results and interpretation

- Relative risk and credible intervals are appropriate outputs, but please avoid causal interpretations. The results should be framed as spatial associations rather than direct effects.

- Highlight that the global autocorrelation results were not significant, and focus more on local clusters.

4. Figures and tables

- Improve figure legends for clarity, especially when reporting cluster categories (HH, HL, LH, LL).

- Tables could be reformatted to improve readability (e.g., Tables 2 and 3).

5. Discussion and limitations

- Expand on limitations of Bayesian smoothing, surveillance-based data, and cross-sectional design.

- Consider linking the findings more explicitly to public health implications, such as targeting vector control or surveillance.

6. Language and style

- The manuscript is written in clear English but would benefit from minor editing to shorten long sentences and improve readability.

Overall recommendation: Major Revision. The manuscript has scientific merit and novelty but requires clarification of methods, stronger discussion of limitations, and improved presentation of results.

Reviewer #11: This is a strong qualitative study that provides nuanced insights into societal views on age-related healthcare prioritization. With minor clarifications and refinements as suggested, the manuscript will make a valuable contribution to the literature on health economics, ethics, and resource allocation.

Reviewer #12: Clarify Terminology Consistently: Ensure consistent use of terms like “children,” “adolescents,” and “young people” throughout the manuscript and figures to avoid confusion.

Improve Figure Quality: Figures 1 and 2 (PTO examples) and Figure 3 (Framework Analysis) should be provided in high-resolution, publication-ready format. Ensure all text is legible.

Reflexivity and Positionality: While limitations are acknowledged, consider briefly expanding on how the interviewers’ backgrounds as economists may have influenced data interpretation, not just data collection.

COREQ Checklist: Ensure the COREQ checklist (S4 Table) is complete and accurately reflects the study’s reporting.

Reference Formatting: Some references (e.g., [23]) are still marked as “unpublished manuscript.” If now published, update citation details accordingly.

Reviewer #13: Congratulations to the author for conducting qualitative research on a very pertinent topic that will guide the policy makers for decision making.

However, there are few suggestions to improve the quality of the research paper.

Please mention the type of qualitative research conducted, what were the strategies used to maintain the rigor of the qualitative study, what method/framework was used for the qualitative data analysis? Moreover, author need to describe in methodology, how many researchers were involved in data collection, how was the inter-coder's reliability data transparency checked? Conclusion could be written aligned to objectives in a comprehensive manner.

All the best for final revisions and publication of the revised manuscript.

Reviewer #14: Concerns from previous reviewers have been addressed adequately, I have just raised minor concerns and add comments that could further improve the document. I support that the manuscript is accepted for publication

**Do you want your identity to be public for this peer review?** For information about this choice, including consent withdrawal, please see our Privacy Policy

Reviewer #4: No

Reviewer #5: **Yes: ** Dr. Ragni Kumari

Reviewer #6: No

Reviewer #7: **Yes: ** Xuanjie Chen

Reviewer #8: **Yes: ** Dr.Justice Ofori-Amoah

Reviewer #9: No

Reviewer #10: No

Reviewer #11: **Yes: ** Dr Neha Asif

Reviewer #12: **Yes: ** Raul Alberto Carrilho Cordeiro

Reviewer #13: **Yes: ** Jarina Begum

Reviewer #14: **Yes: ** Veuvette Ngalulawa Kone

---

## [Author Response · Author response to Decision Letter 2]

8 Oct 2025

Reviewer #4

Dear author,

I understand the importance of evaluating the personal rules involved in choices regarding investments in health programs by age.

1) In your article, I missed the addition of scientific studies on health outcomes related to the age of patients which should be also included to help decision-making. See, for example, articles on investments in health in early childhood. I suggest that you add some references to articles in this sense and that you mention the importance of such a source of decision-making in health.

Response: Thank you for your feedback. We have now incorporated this in the discussion section and added references.

“Seven study participants chose the younger age group because they felt children/adolescents should experience more in life and they should get the opportunity to experience all the things that adults (40 or 55 years) have experienced. This is consistent with the findings from Schweda et al. [32], who reported some participants felt that extending a person’s life was more important and legitimate if they are young because they could experience a more desirable life.

Even though, in the PTO task we explicitly stated that the health gain was the same for both age groups, a few participants did think about the long term benefits a child may get compared to adults. This aligns with scientific evidence supporting long-term benefits of health investments in early childhood (Campbell et al, 2014).

Participants also made their trade off in PTO questions by examining how being in a health condition would affect their families and society as well - a finding also reported by Schweda et al. [32].”

Reference: Campbell F, Conti G, Heckman JJ, Moon SH, Pinto R, Pungello E, et al. Early childhood investments substantially boost adult health. Science. 2014;343(6178):1478-85.

Reviewer #5

Authors addressed all comments raised by the reviewer in beautiful manner. Now this manuscript is ready for publication.

Response: Thank you for your positive feedback on our revised manuscript. We are grateful for your time and effort in reviewing our work.

Reviewer #6

TITLE

1) The title precisely reflects the subject of the study; however, it is academically advantageous to consider substituting the term "people" with "public."

“A qualitative study to understand public views on the relative value of health gains for children and young people compared to adults”

Response: Thank you for raising this point. Reviewer #8 suggested to mention in which country the study was conducted.

Therefore, we have now revised the title as “A qualitative study to understand public views on the relative value of health gains for children and young people in Australia compared to adults”

ABSTRACT

2) Methods: Concise overview. Nonetheless, I suggest the inclusion of the following statement at the last line of the methods used paragraph: “Thematic analysis was employed to identify fundamental reasoning patterns.”

Response: Thank you for the comment. We have now added the sentence into the methods section.

“Methods: Think-aloud, semi-structured interviews were conducted with Australian adolescents (n=7), non-parents (n=11), parents with healthy children (n=8) and parents of children with health conditions (n=15). Participants completed Person Trade-Off (PTO) and attitudinal questions about resource allocation for improvements in life extension, mental health, mobility, and pain/discomfort choosing between interventions for adults (ages 40 or 55) and younger people (ages one month to 24). Thematic analysis was employed to identify fundamental reasoning patterns.”

INTRODUCTION

The introduction is thorough and based in robust research. However, Structural enhancements are necessary to improve the clarity and scholarly tone, such as:

3) Lines (108 –128), (129-142), The paragraphs are too dense and long, consider break them into clear paragraphs.

Response: Thank you for raising this point. We agree that dividing the paragraphs will enhance the readability. The section now reads as follows:

“Differences in the age-related social values of QALYs are usually based on one of two rationales, efficiency considerations or equity/fairness considerations [3]. Efficiency considerations draw on the relationship between economic productivity and age, with consideration for broader social contributions to vary by age [4]. Fairness considerations draw on the fair innings argument. This was first presented by Harris [5] who argues that everyone should be given an equal opportunity to reach a normal span of years. According to this view, those who have not had their fair innings need to be prioritized. Williams [6] extended the fair innings argument to incorporate a person’s quality of life as well as length of life. Tsuchiya et al. [7] reported that people favour giving priority to younger people based on the fair innings argument, and to older people based on efficiency considerations. Schwappach [8] suggested the social value of a QALY may vary according to patients’ characteristics i.e., age, social role, lifestyle or severity of illness.

Existing studies have explored whether age should be considered as a criterion in allocating healthcare resources but the evidence on preference relating to age-related prioritization is mixed. Some studies identified a willingness to prioritize all children aged below 15 [9]. Other studies suggest support for prioritising children but only provide evidence for those older than 5 years. For example, Richardson et al. [10] reported age weights for ages from 5 to 70. They reported a preference for age 5, 10, 15 and 20 for life extension and age 5, 10, 15 for quality of life improvements. Petrou et al. [11] reported relative age weights from a person trade-off (PTO) study and identified a preference for prioritizing the younger age for life extending treatments. In contrast, there are also studies which found participants prioritize adults aged 40 and 70 years over children aged 10 years [12]. Some studies produce findings suggesting everyone should be equally treated [13]. A few qualitative studies analyzed people’s views on prioritizing treatments for children compared to adults. Aidem [14] reported that policy makers believe healthcare needs to be prioritized based on efficiency and equity. Kuder and Roeder [15] reported that people believe patients should not be treated differently based on their age.

A recent systematic review synthesised international evidence on the relative social value of health gains for children (<18 years) and those of adults [16]. The review found evidence that the public were willing to prioritize children’s health gains over adult’s health gains. However, the review identified variations in results (1) based on the study methodology. For example the review identified differences in results based on the type of question i.e., attitudinal questions [17, 18] compared to choice based numerical questions [19], (2) across different perspectives the study questions were framed i.e., prioritization within the family, or as a citizen or adopting a decision maker perspective, (3) based on the age of the child, (4) based on participant characteristics such as age, gender, parental status and (5) based on whether the health gain referred to extensions of length of life or improvements in quality of life.

One of the limitations identified in this review was the limited number of studies that explored the rationale behind participant choices. Tsuchiya [20] is a rare example. Therefore, further research is needed to understand what drives individuals’ responses to preference elicitation questions in which age of the recipient of health gain differs. Qualitative work can help interpret such variations in findings, by providing an understanding of the underlying reasons, and insights into participants’ thinking patterns and principles when responding to different types of questions [14, 15].”

4) Line 113, those who have not had their fair innings needs to be prioritized. Change needs to “need”

“According to this view, those who have not had their fair innings need to be prioritized.”

5) Line 120, consider changing >5 years to “older than 5 years” for more academic tone.

“Other studies suggest support for prioritising children but only provide evidence for those older than 5 years”

METHODS

Demonstrate methodological transparency, provide a strong rationale for recruitment, and offer clear justification for the selections. Nevertheless, it requires slight modifications for improved clarity, formatting, and organization.

6) Recruitment and Participants: Lines (164-179) and (180-192) consist of extensive paragraphs; please divide them into shorter paragraphs.

Response: Thank you for your suggestion on improving readability of the paper. We have now divided this paragraph into sections and also have made a change to the sentence involving adolescents to highlight reviewer #8 suggestions.

“Existing literature has found that attitudes towards prioritizing child health gains vary depending on the age and parenthood status of the participants [16]. In addition to age and parenthood, we hypothesized that parent’s experiences of child ill health may be relevant. Therefore, our recruitment ensured coverage of non-parents, parents of healthy children, and parents of children with a health condition across different age groups.

We also included older adolescents (aged 16 -18 years) at the request of the QUOKKA’s Decision Makers’ Panel. The recent review [16] identified that there was a lack of qualitative studies specifically examining adolescent’s perspective on age-based healthcare prioritization, yet studies have shown it is feasible for adolescents to value health states [24] which involves tasks with a similar level of cognitive and emotional difficulty to PTO questions. Adolescents (aged 16-18) coped well during the pilot interviews, which are described at length elsewhere [22].

We anticipated saturation would be reached at a sample of 40 interviews on the basis of Ritchie et al. [25] who suggested that studies involving a very diverse population might require an increased sample size, but a sample of fewer than 50 will be adequate for individual interviews. This is further supported by Hennink et al. [26] who suggested saturation would be achieved between 16-24 interviews and Guest et al. [27] who found saturation occurred within 12 interviews. Consideration of saturation adopted the approach by Guest et al. [27] which “refers to the point during data analysis at which incoming data points (interviews) produce little or no new useful information relative to the study objectives” [27].”

7) Survey Design: The paragraph is extensive, consisting of lines 198-210. It is desirable to divide it into two paragraphs.

Response: Thank you for your suggestion. The survey design section now reads as follows:

“The survey included six components, including consent and introduction video, seven PTO questions, feedback questions on comprehension, questions asking for reasons for PTO answers, attitudinal questions on health prioritization and demographic questions. Further details of the tasks and questions are provided in the published protocol study [22]. The seven PTO tasks involved different aspects of health improvement, including life extension (2 or 5 years), and improvements in aspects of quality of life (mental health, mobility and pain or discomfort). Examples are shown in Figure 1 and 2. There were four life extension questions and three quality of life questions.

Participants were asked to make choices between pairs of interventions, one impacting an adult group (either 40 or 55 years old) and the other younger group consisting of 13 age categories (one month, even number of years between 1 year and 24 years old). One of the life extension questions (applied to all participants) compared young people to other young people of a different age as part of a chaining test for the quantitative study. The ages used in the PTO questions was randomly selected, however for the final four interviews an age < 4 years was chosen for the younger age group to further explore findings arising from the analysis of the main survey data. Half of the sample were randomly given the option to select ‘no preference’ (unforced-arm) between the two hypothetical health programs in the seven PTO tasks, the other half were always required to choose between Program A or B (forced-arm). Interviewer prompts included discussion of their likely answer if they had seen the alternative presentation (i.e., when a choice between two programs was forced).”

8) Data Collection: The paragraphs lines (218-227) and (233-253) are extensive; therefore, it is necessary to divide them into separate paragraphs.

Response: Thank you for your suggestion. The data collection section now reads as follows:

• Line 218-227

“The interviews were conducted by three female interviewers (AD, TPE and CB). AD had experience in conducting quantitative interviews with adults and received training at the start of the study. TPE and CB were experienced in conducting qualitative interviews. Regular debriefs following the interviews were conducted among the three interviewers. Discussions included reflecting on initial interviews, re-listening to interview recordings and evaluating interviewer prompts.

We conducted an initial two pilot interviews with a convenience sample (known to the interviewers) to confirm interview prompts and processes; this data was not included in the analysis. The first six interviews were treated as a second pilot. The pilot interviews were rewatched and discussed including a reflection on the whether the prompts led to discussion which addressed the study questions. As no major changes were made to the interview prompts at this stage this pilot data is included in the main sample. Detailed information about the pilot and development of the survey prompts are described in the protocol paper [22].”

• Line 233-253

“At the start of the interview the interviewer explained how to participate in a think-aloud interview by demonstrating an example question. During the interview if the participants became quiet, they were encouraged to think-aloud and to explain why they had made their choice. The interviewers then completed the survey on behalf of the participants.

We acknowledge the potential bias from using interviewer-led mode. However, interviewer-led mode could help improve participant’s comprehension of the survey tasks and facilitate responses. We had made several efforts to mitigate potential biases, first, using neutral interview prompts (e.g., “I would like you to talk me through your thought process – just saying whatever is coming into your head. There are no right or wrong answers or thoughts here”) to ask survey questions, second, the survey began with an introduction video which talked through an example PTO question. In this video it was mentioned that participants could have views favoring young people or favoring adults.

Participants then answered seven PTO questions and three attitudinal questions and were invited to ‘think-aloud’ while they answered and to provide reasons behind their responses. They were also asked semi-structured questions to further probe their thinking and the reasons for their answers by answering feedback questions. The interviewers probed to explore differences in PTO responses between the different types of health gain. Where PTO responses appeared to give different preferences to attitudinal responses the interviewer asked for an explanation. If appropriate, the interviewer referred to other participants who had given apparently inconsistent views on these questions to ensure the participant did not feel their responses were being challenged. To explore the strength and robustness of the participants’ views the interviewer also raised that some people had given us very different opinions to their own (e.g. prioritizing children over adults or prioritizing adults over children) and asked how they felt about this. lnterviews were video recorded and transcribed intelligent verbatim using automated transcription which was checked by interviewers using the video.”

RESULTS

9) Participants secti

---

## [Decision Letter · Decision Letter 2]

21 Oct 2025

A qualitative study to understand public views on the relative value of health gains for children and young people in Australia compared to adults

PONE-D-25-04838R2

Dear Dr. De Silva,

We’re pleased to inform you that your manuscript has been judged scientifically suitable for publication and will be formally accepted for publication once it meets all outstanding technical requirements.

Kind regards,

Paolo Landa, Ph.D.

Academic Editor

PLOS ONE

Additional Editor Comments (optional):

The authors well answered to the reviewers and improved the manuscript considering the quality standards of the journal. Congratulation to the authors for the great result.

Reviewers' comments:

Reviewer's Responses to Questions

**Comments to the Author**

Reviewer #5: All comments have been addressed

Reviewer #7: All comments have been addressed

Reviewer #8: All comments have been addressed

Reviewer #12: All comments have been addressed

2. Is the manuscript technically sound, and do the data support the conclusions?

Reviewer #5: Yes

Reviewer #7: Yes

Reviewer #8: Yes

Reviewer #12: Yes

3. Has the statistical analysis been performed appropriately and rigorously?

Reviewer #5: (No Response)

Reviewer #7: Yes

Reviewer #8: Yes

Reviewer #12: Yes

4. Have the authors made all data underlying the findings in their manuscript fully available?

Reviewer #5: Yes

Reviewer #7: Yes

Reviewer #8: Yes

Reviewer #12: Yes

5. Is the manuscript presented in an intelligible fashion and written in standard English?

Reviewer #5: Yes

Reviewer #7: Yes

Reviewer #8: Yes

Reviewer #12: Yes

Reviewer #5: Thank you for the opportunity to review this revised manuscript. The authors have satisfactorily addressed previous reviewer comments, including expanding the literature on age-related health outcomes, and clarifying their qualitative methodology and results interpretation.

The study is technically sound, uses appropriate methods, and the data support the conclusions well. Ethical standards are upheld, and data availability is adequately described. The manuscript is clearly written in standard English and is presented intelligibly for the journal readers.

I recommend acceptance of this manuscript for publication in PLOS ONE. It provides original, valuable insights into public preferences on age-based health prioritization and will contribute meaningfully to healthcare resource allocation literature.

Reviewer #7: (No Response)

Reviewer #8: Errors and typos identified are rectified by the authors,given the large study area,study participants and the objectives of the study the authors can consider getting more publications

Reviewer #12: No further comments. Main comments have been adressed before. The main recomendations from all rewieres have been considered.

**Do you want your identity to be public for this peer review?** For information about this choice, including consent withdrawal, please see our Privacy Policy

Reviewer #5: **Yes: ** Dr. Ragni kumari

Reviewer #7: **Yes: ** Xuanjie Chen

Reviewer #8: **Yes: ** Justice Ofori-Amoah

Reviewer #12: **Yes: ** Raul Alberto Cordeiro

---

## [Editor Report · Acceptance letter]

PONE-D-25-04838R2

PLOS ONE

Dear Dr. De Silva,

I'm pleased to inform you that your manuscript has been deemed suitable for publication in PLOS ONE. Congratulations! Your manuscript is now being handed over to our production team.

Kind regards,

on behalf of

Dr. Paolo Landa

Academic Editor

PLOS ONE